# LinearRAG: Linear Graph Retrieval Augmented Generation on Large-scale Corpora

**Luyao Zhuang**[1][*] **Shengyuan Chen**[1][*]**, Yilin Xiao**[1]**, Huachi Zhou**[1][†]**, Yujing Zhang**[1]
**Hao Chen**[1]**, Qinggang Zhang**[1][†]**, Xiao Huang**[1]
[1]The Department of Computing, Hong Kong Polytechnic University, Hong Kong SAR
```
{luyao.zhuang,yilin.xiao,yu-jing.zhang}@connect.polyu.hk;
huachi.zhou@connect.polyu.hk; sundaychenhao@gmail.com;
{sheng-yuan.chen,qinggang.zhang,xiao.huang}@polyu.edu.hk †
```

## ABSTRACT

Retrieval-Augmented Generation (RAG) is widely used to mitigate hallucinations of Large Language Models (LLMs) by leveraging external knowledge. While effective for simple queries, traditional RAG systems struggle with large-scale, unstructured corpora where information is fragmented. Recent advances incorporate knowledge graphs to capture relational structures, enabling more comprehensive retrieval for complex, multi-hop reasoning tasks. However, existing graph-based RAG (GraphRAG) methods rely on unstable and costly relation extraction for graph construction, often producing noisy graphs with incorrect or inconsistent relations that degrade retrieval quality. In this paper, we revisit the pipeline of existing GraphRAG systems and propose **Linear** Graph-based **R**etrieval-**A**ugmented **G**eneration (**LinearRAG**), an efficient framework that enables reliable graph construction and precise passage retrieval. Specifically, LinearRAG constructs a relation-free hierarchical graph, termed Tri-Graph, using only lightweight entity extraction and semantic linking, avoiding unstable relation modeling. This new paradigm of graph construction scales linearly with corpus size and incurs no extra token consumption, providing an economical and reliable indexing of the original passages. For retrieval, LinearRAG adopts a two-stage strategy: (i) relevant entity activation via local semantic bridging, followed by (ii) passage retrieval through global importance aggregation. Extensive experiments on four benchmark datasets demonstrate that LinearRAG significantly outperforms baseline models. Our code and datasets are available at https://github.com/DEEP-PolyU/LinearRAG.

## 1 INTRODUCTION

Retrieval-Augmented Generation (RAG) has emerged as a promising approach to enhance Large Language Models (LLMs) by leveraging external knowledge bases (Gao et al., 2023; Zhou et al., 2025a). However, existing RAG systems struggle with the complexities of large-scale, unstructured corpora in real-world scenarios, where the relevant information is frequently distributed unevenly across heterogeneous documents. The context retrieved by RAG systems is often voluminous, intricate, and lacks clear organization, leading to issues of variability in accuracy and coherence (Sun et al., 2024; Zhang et al., 2024). Although recent advances attempt to manage this by segmenting documents into smaller chunks for efficient indexing (Borgeaud et al., 2022; Izacard et al., 2023; Jiang et al., 2023), this strategy often results in the loss of critical contextual details, impairing retrieval accuracy and reasoning capabilities for complex tasks (Han et al., 2024; Zhang et al., 2025b).

To address this, Graph Retrieval-Augmented Generation (GraphRAG) (Zhang et al., 2025b; Procko & Ochoa, 2024; Xiao et al., 2025a) has recently emerged as a powerful paradigm that leverages external structured graphs to model the hierarchical structure of background knowledge (Han et al., 2024). Specifically, early work, like RAPTOR (Sarthi et al., 2024) and Microsoft's

---

[*]Equal contribution.
[†]Corresponding author: Qinggang Zhang, Huachi Zhou.

GraphRAG (Edge et al., 2024), organizes knowledge through recursive summarization and community detection with LLM-generated synopses, enabling coarse-to-fine retrieval for comprehensive responses. Building on this, recent approaches, including GFM-RAG (Luo et al., 2025a), G-Retriever (He et al., 2024), and LightRAG (Guo et al., 2024), integrate specialized encoders and objectives, such as query-dependent GNNs, Prize Collecting Steiner Trees, and dual-level indexing, to improve multi-hop generalization, scalability, and efficiency. More recently, HippoRAG (Gutiérrez et al., 2024) and its enhancement HippoRAG2 (Gutiérrez et al., 2025) draw inspiration from cognitive processes to utilize personalized PageRank for multi-hop retrieval. These strategies significantly improve retrieval precision and contextual depth, enabling LLMs to address complex, multi-hop queries more effectively.

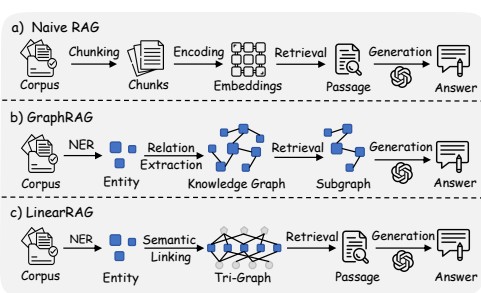

Figure 1: Three paradigms of RAG systems.

Despite its conceptual promise and theoretical superiority, recent studies reveal that GraphRAG models frequently underperform even naive RAG approaches on many real-world applications (Han et al., 2025; Zhou et al., 2025c; Zhuang et al., 2025). This performance degradation mainly stems from the poor quality of automatically constructed knowledge graphs. While graph-based retrieval increases recall of relevant knowledge, it concurrently introduces substantial noise and ambiguities into the retrieved contexts, due to errors in graph construction. Specifically, two critical deficiencies undermine graph quality, including (i) local inaccuracy: relation extraction processes exhibit significant error rates, resulting in inaccurate semantic relationships between entities. (ii) global inconsistency: the absence of mechanisms to enforce hierarchical consistency and global coherence during extraction leads to structurally fragmented graphs with poor connectivity. These deficiencies collectively manifest as structural conflicts and semantic ambiguity within the knowledge graph, which subsequently corrupt the retrieval and generation processes. Although recent attempts have been made to refine graph quality via bottom-up clustering-based community summarization (Edge et al., 2024; Gutiérrez et al., 2025; Wang et al., 2025) or topic modeling (Sarthi et al., 2024) to offer a broader, macro-level view of data, these unsupervised methods are vulnerable to error propagation, where inaccuracies in entity relationships are amplified at higher levels of abstraction.

In this paper, we revisit the pipeline of existing GraphRAG systems and propose **Linear** Graph-based **R**etrieval-**A**ugmented **G**eneration (**LinearRAG**), a framework that enables efficient, reliable graph construction and precise corpus retrieval with multi-hop reasoning. The core idea of LinearRAG is to simplify the complex relational graph into a linear, easy-to-index view by focusing solely on modeling the semantics between target entities and the underlying text passages. Instead of relying on costly relation extraction, LinearRAG constructs a hierarchical graph from entities, sentences, and passages, using only lightweight entity extraction and semantic linking. On top of this graph, LinearRAG introduces a two-stage passage retrieval technique: ❶ **local semantic bridging for entity activation**, which identifies contextually relevant entities beyond literal matches by propagating semantic similarity in sentences to mine multihop contextual association; and ❷ **global importance aggregation for passage retrieval**, which applies personalized PageRank over the activated subgraph to aggregate passage importance from a holistic perspective. Together, these modules enable LinearRAG to achieve scalable, accurate, and noise-resilient retrieval for complex queries. Our overall contributions are summarized as follows:

- We identify key limitations in existing GraphRAG systems, specifically highlighting how reliance on unstable relation extraction introduces noise and structural inconsistencies. This motivates the design of LinearRAG, a novel framework that enables reliable graph construction and precise passage retrieval while maintaining linear scalability.

- LinearRAG constructs a relation-free hierarchical graph, called Tri-Graph, using only lightweight entity extraction and semantic linking, which avoids the instability of traditional relation modeling and reduces indexing time by over 77%.

- On top of the constructed graph, we design a two-stage retrieval mechanism that combines local semantic bridging for precise entity activation with global importance aggregation for

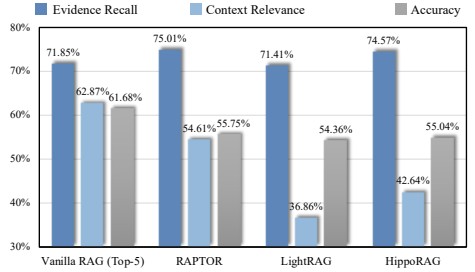
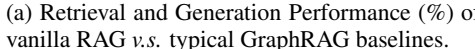

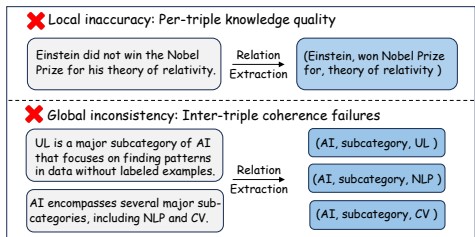

(a) Retrieval and Generation Performance (%) of vanilla RAG *v.s.* typical GraphRAG baselines.

(b) Two types of errors in knowledge graphs brought by imperfect relation extraction methods.

Figure 2: **(a) Retrieval and generation rerformance (%) of Vanilla RAG *v.s.* GraphRAG Baselines.** Notably, the evaluation on Medical dataset measures GPT-based accuracy, context relevance, and evidence recall across different RAG baselines. **(b) Case study of relation errors in knowledge graph construction** from local inaccuracy and global inconsistency perspectives.

> passage recall. This integrated strategy enables more accurate, noise-resilient, and single-pass multi-hop retrieval.

- We conduct extensive experiments on four benchmark datasets, demonstrating that LinearRAG consistently outperforms state-of-the-art baselines in terms of retrieval quality, generation accuracy, and scalability, validating its practicality for real-world applications.

## 2 PRELIMINARY STUDY

In this section, we conducted a series of preliminary studies to investigate the effect of graphs used in RAG systems. Our findings reveal critical flaws in graph construction that explain the underlying causes of GraphRAG's frequent underperformance compared to naive RAG in traditional tasks.

### 2.1 PERFORMANCE DEGRADATION IN GRAPHRAG SYSTEMS

GraphRAG models frequently underperform traditional RAG approaches on many real-world tasks. Specifically, our experiments on GraphRAG-Bench (Xiang et al., 2025) demonstrate that while GraphRAG leverages graph structures to enhance recall by retrieving a wider array of potentially relevant passages, this gain is offset by a significant introduction of noisy and ambiguous contextual information. Specifically, GraphRAG methods such as LightRAG and HippoRAG achieve moderate improvements in evidence recall, but exhibit significantly lower context relevance, ranging from 36.86% to 54.61%, compared to Vanilla RAG, which attains 62.87% performance. This suggests that although graph-based retrieval expands the scope of contextual information, it introduces substantial noise that compromises the relevance and reliability of generated answers. For example, in a question-answering task about "climate change impacts", GraphRAG might retrieve passages related to "economic policies" due to tenuous graph links. In contrast, vanilla RAG preserves tighter alignment with the query context, leading to more accurate and stable outputs.

### 2.2 GRAPH QUALITY AND ERROR ANALYSIS

To diagnose the root cause of performance degradation observed in Figure 2a, we performed a fine-grained error analysis on the knowledge graphs used in GraphRAG. Our analysis revealed that this performance degradation stems directly from deficiencies in knowledge graph construction. Traditional GraphRAG pipelines rely on explicit relation extraction to construct relational graphs, which usually introduce errors at two levels: (i) Local Inaccuracies: relation extraction models often produce factually incorrect triples. For example, as shown in Figure 2b, the sentence "Einstein did not win the Nobel Prize for his theory of relativity" may be misrepresented as (Einstein, won Nobel Prize for, theory of relativity), fundamentally altering factual meaning. (ii) Global Inconsistencies: existing relation extraction is performed locally on individual text passages, with no mechanism to validate or reconcile connections across the entire corpus, leading to redundant or contradictory relations. For example, "AI" may be linked to "Unsupervised Learning," "NLP," and "CV" as parallel

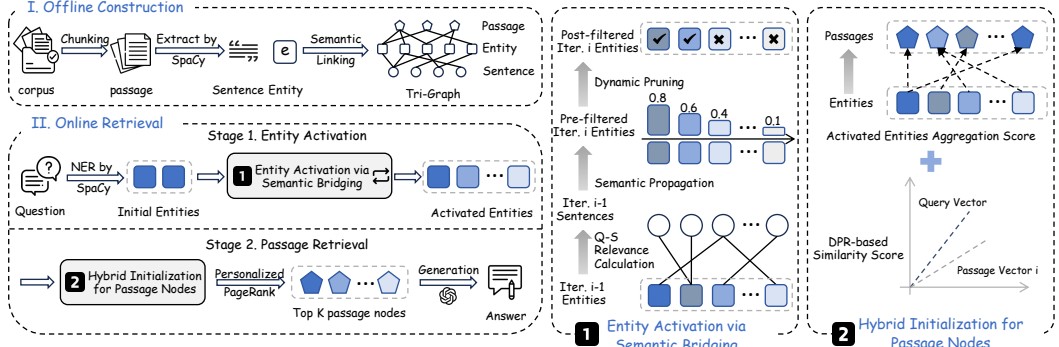

Figure 3: **The overall pipeline of the proposed LinearRAG framework. I. Offline Construction.** Initially, we construct a Tri-graph containing entity, sentence, and passage nodes, with edges connecting entities to sentences and entities to passages. **II. Online Retrieval.** We first activate relevant entities via local semantic bridging on the entity-sentence subgraph while fixing passage nodes, then using the activated entities to aggregate global importance scores, finally, perform passage retrieval via personalized PageRank on the entity-passage subgraph while fixing sentence nodes.

subcategories without hierarchical coherence (e.g., that NLP and CV are subfields of AI, while "Unsupervised Learning" is a technique used within them). This structural ambiguity directly misleads the retrieval process, introducing semantic noise based on these inconsistent connections.

## 2.3 DISCUSSION

The conventional GraphRAG pipeline relies heavily on explicit relation extraction and triple-based knowledge representation. While this approach aims to summarize passages into structured relational forms, it faces two fundamental issues: **First, extracting concise and accurate relational triples is computationally expensive and linguistically challenging.** Relations expressed in natural language are often complex and context-dependent, and often nuanced or compositional to be accurately distilled into atomic triples; for example, the sentence "Rachel reluctantly agreed to go running with Phoebe" cannot be cleanly reduced to a single atomic triple without losing critical semantic nuance. **Second, explicit relation extraction is unnecessary.** Aligned entities, rather than relations, serve as the primary anchors connecting information distributed across passages. The original text preserves relational semantics in full context, which can be interpreted dynamically by large language models during inference, without relying on error-prone extraction.

## 3 THE FRAMEWORK OF LINEARRAG

Preliminary findings indicate that explicit relation extraction is not only computationally expensive but also largely unnecessary. These insights inform our revisit of the design of GraphRAG components, culminating in two central claims: (i) Aligned entities serve as the primary anchors connecting information distributed across passages. (ii) Contextual relations are best preserved within the original passages, eliminating the need for explicit relation extraction.

Motivated by this idea, we introduce a new GraphRAG paradigm as in Figure 3. It constructs a graph that contains three types of nodes: entity nodes, sentence nodes, and passage nodes. Edges connect entities to sentences and entities to passages, represented as two adjacency matrices. Retrieval proceeds in two stages: ❶ **entity activation stage**, where we fix the passage nodes and use local semantic bridging on the entity–sentence subgraph to identify intermediate entities that connect different passages; and ❷ **passage retrieval stage**, where we fix the sentence nodes and apply personalized PageRank on the entity–passage subgraph, exploiting the activated entities in the first stage as seeds for global importance aggregation. This framework exhibits linear scalability in both graph construction and retrieval (detailed analysis in Appendix D), and we refer to it as **LinearRAG**.

## 3.1 Token-free Graph Construction

We construct a hierarchical graph, called Tri-graph, with multiple granularities that is efficient to maintain and update. Given a corpus with a set of passages $\mathcal{P}$, we first segment each passage into sentences using punctuation (e.g., periods or exclamation marks), obtaining a sentence set $\mathcal{S}$. We then apply lightweight models (e.g., spaCy (Honnibal et al., 2020)) for named entity recognition (NER) to derive an entity set $\mathcal{E}$. The passages, sentences, and entities constitute three types of nodes in the graph, denoted as $V_p$, $V_s$, and $V_e$, respectively.

Edges are constructed according to the following rules: if a passage $p_i$ contains an entity $e_j$, we add an edge $(V_{p_i}, V_{e_j})$; likewise, if a sentence $s_i$ mentions an entity $e_j$, we add an edge $(V_{s_i}, V_{e_j})$. These relations correspond to two adjacency matrices: the *contain matrix* $C$ between passages and entities, and the *mention matrix* $M$ between sentences and entities.

Formally, the contain matrix $C$ is defined as an $|V_p| \times |V_e|$ matrix:

$$C = [C_{ij}]_{|V_p| \times |V_e|}, \quad \text{where} \quad C_{ij} = \mathbb{1}_{\{p_i \text{ contains } e_j\}}. \tag{1}$$

Here, $\mathbb{1}$ denotes the indicator function, with $C_{ij} = 1$ if $p_i$ contains $e_j$, and $C_{ij} = 0$ otherwise.

Similarly, the mention matrix $M$ is defined as an $|V_s| \times |V_e|$ matrix:

$$M = [M_{ij}]_{|V_s| \times |V_e|}, \quad \text{where} \quad M_{ij} = \mathbb{1}_{\{s_i \text{ mentions } e_j\}}. \tag{2}$$

This design enables efficient graph updates as the corpus grows. When new passages arrive, only those passages undergo sentence segmentation, NER, and edge construction, yielding overall linear complexity. Notably, NER is both precise and efficient compared with OpenIE, and can be performed using lightweight language models (e.g., BERT-based models in spaCy) without incurring LLM token costs. In addition, the adjacency matrices $C$ and $M$ are implemented in sparse form, exploiting their intrinsic sparsity to further reduce memory usage to a linear-scalability. Finally, by retaining the original passages as knowledge carriers, the resulting graph preserves all contextual information, thereby ensuring information-lossless construction.

## 3.2 Passage Retrieval

With the constructed graphs serving as lossless knowledge carriers, the objective becomes identifying the most informative passages effectively. The design of graph-based retrieval is critical, as it must balance precision and recall of relevant context, particularly in the case of multihop queries. We decompose the retrieval process into two stages: a precise entity activation stage via local semantic bridging followed by a passage retrieval stage to recall globally important passages.

### 3.2.1 First Stage: Relevant Entity Activation via Semantic Bridging

Generally, direct entity matching can miss relevant intermediate entities that bridge multihop relations, which are essential for multihop queries. Therefore, it is crucial to identify these latent connectors. However, such intermediate entities cannot be directly mined, and the massive number of entities challenges the precision of identifying them. To tackle this, we propose *relevant entity activation via semantic bridging*. For an incoming query $q$, the process is as follows:

**Initial entity activation** that identifies entities contained in the query $q$ and represents their activation in the knowledge graph as a sparse vector. Specifically, we first use SpaCy to extract entities from $q$, denoted as $E_q$. Then for each extracted entity, we find the most similar entity in the knowledge graph. The initial activation score of the matched entity is set to its similarity score:

$$\mathbf{a}_q = [\mathbf{a}_{q,i}]_{|V_e| \times 1}, \quad \text{where} \quad \mathbf{a}_{q,i} = \mathbb{1}_{i = \arg\max_{e_j \in V_e} \text{sim}(e_q, e_j)} \cdot \text{sim}(e_q, e_i). \tag{3}$$

**Query-sentence relevance distribution** that computes the contextual association between the query $q$ and each sentence $s_i \in S$, where $S = \{s_1, s_2, \ldots, s_{|S|}\}$ is the set of sentences in the corpus. Represent these similarities as a vector:

$$\sigma_q = [\sigma_{q,i}]_{|S| \times 1}, \quad \text{where} \quad \sigma_{q,i} = \text{sim}(q, s_i). \tag{4}$$

**Semantic propagation** that propagates the similarities in $\sigma_q$ through semantic similarity extraction to activate relevant intermediate entities, enabling the bridging of multihop relations in the

knowledge graph. The entity activation vector $\mathbf{a}_q$ is updated with the weighted aggregation of their associated neighbors in the sentence-entity bipartite graph using the matrix $M$ from equation 2:

$$\mathbf{a}_q^t = \texttt{MAX}(M^T(\sigma_q \odot (M\mathbf{a}_q^{t-1})), \mathbf{a}_q^{t-1}), \tag{5}$$

where $\mathbf{a}_q^t$ is the activation vector of the entities in the $t_{th}$ iteration of semantic propagation. With a few iterations, we identify a set of contextually relevant entities that anchors a subgraph in the corpus that aligns with the reasoning structure of the query. This solution mimics the *relation-matching* process in the GraphRAG algorithms for multi-hop reasoning. Differently, our strategy performs implicit relation matching and does not rely on explicitly constructed relational knowledge graph.

Noteworthy, the vectorized formulation of equation 5 enables $n$-hop entity activation within only $n$ iterations, where $n$ is generally small ($\leq 4$). Each iteration involves two $\texttt{MatMul}$ (matrix multiplication) and one $\texttt{MAX}$ operation, which can be efficiently implemented and executed in parallel on popular machine learning platforms. Furthermore, due to the intrinsic sparsity of the matrix $M$ and the activation vector $\mathbf{a}_q^t$, we can store it in a sparse format to reduce memory consumption, and reduce computation cost by applying $\texttt{SpMM}$ (sparse matrix multiplication) to replace $\texttt{MatMul}$.

**Dynamic pruning.** While the above semantic bridging successfully establishes initial associations from query entities to potentially relevant intermediates, it also introduces a major challenge: the exponential growth of the search space as propagation deepens. Without proper control, irrelevant entities may repeatedly serve as new seeds, causing the process to expand combinatorially and drift into semantic regions unrelated to the original query intent. To address this issue, we perform graph pruning that constrains expansion to high-quality semantic paths. Specifically, at each propagation step, we introduce a threshold $\delta$. A newly activated entity is retained for the next iteration only if its relevance score exceeds $\delta$; otherwise, it is pruned. Moreover, the process terminates automatically once no new entity surpasses the threshold. This adaptive mechanism ensures that propagation proceeds only along the most relevant semantic paths, while dynamically tailoring the iteration range to the complexity of each query.

### 3.2.2 SECOND STAGE: PASSAGE RETRIEVAL VIA GLOBAL IMPORTANCE AGGREGATION

In the first stage, relevant entities are extracted via local semantic-bridged subgraph expansion in the sentence-entity graph. These entities are used to initialize importance scores in a passage-entity graph, a bipartite graph where nodes represent passages ($V_p$) and entities ($V_e$), with edges connecting passages to their contained entities based on occurrence. The second stage performs global importance aggregation with hybrid initialization for passage nodes to retrieve important passages using Personalized PageRank (PPR) on the passage-entity graph to compute refined global importance scores for each node $v_i \in V_p \cup V_e$:

$$I(v_i) = (1 - d) + d \cdot \sum_{v_j \in B(v_i)} \frac{I(v_j)}{\deg(v_j)} \tag{6}$$

Here, $d$ is the damping factor (typically 0.85), $B(v_i)$ is the set of nodes linking to $v_i$, and $\deg(v_j)$ is the number of outgoing links from node $v_j$. The initial importance score for entity node $v_i \in V_e$ is set to $I(v_i | v_i \in V_e) = \mathbf{a}_q^{(i)}$, where $\mathbf{a}_q = (\mathbf{a}_q^{(1)}, \mathbf{a}_q^{(2)}, \dots)$ is a vector of entity relevance scores for query $q$ computed in the first stage. For passage nodes $v \in V_p$, the initial importance score is:

$$I(v | v \in V_p) = \left( \lambda \cdot \texttt{sim}(q, v) + \ln\left( 1 + \sum_{e_i \in E_a} \frac{\mathbf{a}_q^{(i)} \cdot \ln(1 + N_{e_i})}{L_{e_i}} \right) \right) \cdot W_p, \tag{7}$$

where $\texttt{sim}(q, v)$ is the PPR base score capturing similarity between passage $v$ and query $q$, $E_a$ is the set of activated entities from the first stage, $N_{e_i}$ is the occurrence count of entity $e_i$ in passage $v$, $L_{e_i}$ is the hierarchical level of entity $e_i$, and $W_p$ is the passage node weight coefficient, $\lambda$ is a trade-off coefficient. Passages are ranked by their PPR scores $I(v | v \in V_p)$, and the top-$k$ passages with the highest scores are selected for retrieval.

## 4 EXPERIMENTS

In this section, we conduct comprehensive experiments to verify the effectiveness and efficiency of LinearRAG. Specifically, we aim to answer the following questions. **Q1** (Generation Accuracy):

Table 1: **Result (%) of baselines and LinearRAG on four benchmark datasets in terms of both Contain-Match and GPT-Evaluation Accuracy.** The best result for each dataset is highlighted in **bold**, while the second result is indicated with an underline.

| Method | HotpotQA | | 2Wiki | | MuSiQue | | Medical |
|---|---|---|---|---|---|---|---|
| | Contain-Acc. | GPT-Acc. | Contain-Acc. | GPT-Acc. | Contain-Acc. | GPT-Acc. | GPT-Acc. |
| *Direct Zero-shot LLM Inference* | | | | | | | |
| llama-8B | 31.10 | 27.30 | 33.60 | 16.20 | 7.40 | 8.10 | 27.31 |
| llama-13B | 24.20 | 16.80 | 21.90 | 10.50 | 3.30 | 4.40 | 28.86 |
| GPT-3.5-turbo | 33.40 | 43.20 | 28.70 | 31.00 | 10.30 | 21.90 | 45.60 |
| GPT-4o-mini | 38.90 | 40.20 | 36.30 | 31.40 | 13.60 | 15.80 | 42.10 |
| *Vanilla Retrieval-Augmented-Generation* | | | | | | | |
| Retrieval (Top-1) | 46.30 | 49.10 | 36.60 | 31.70 | 17.80 | 21.10 | 48.01 |
| Retrieval (Top-3) | 53.00 | 56.00 | 44.90 | 39.70 | 25.10 | 27.50 | 59.07 |
| Retrieval (Top-5) | 55.70 | 58.60 | 48.60 | 43.00 | 26.10 | 29.60 | 61.68 |
| RerankRAG | 55.90 | 59.60 | 49.30 | 45.30 | 26.70 | 31.80 | 55.43 |
| RCOT | 56.30 | 60.60 | 49.80 | 45.30 | 26.80 | 30.70 | 48.69 |
| *Graph-based Retrieval-Augmented-Generation Methods* | | | | | | | |
| KGP | 61.50 | 60.90 | 31.60 | 30.00 | 25.60 | 30.10 | 54.22 |
| G-retriever | 42.20 | 40.60 | 46.60 | 27.10 | 14.40 | 15.50 | 50.36 |
| RAPTOR | 55.90 | 58.30 | 50.10 | 42.10 | 23.30 | 27.40 | 55.75 |
| $E^2$GraphRAG | 61.00 | 63.90 | 54.30 | 38.10 | 23.80 | 26.20 | 58.00 |
| LightRAG | 60.30 | 59.50 | 55.20 | 39.00 | 27.40 | 28.60 | 54.36 |
| HippoRAG | 57.00 | 59.30 | 66.10 | 59.90 | 29.30 | 24.10 | 55.04 |
| GFM-RAG | 62.70 | 65.60 | 66.80 | 59.50 | 29.90 | 34.60 | 56.07 |
| HippoRAG2 | 62.90 | 64.30 | 62.70 | 55.00 | 31.00 | 35.00 | 60.77 |
| **LinearRAG (Ours)** | **64.30** | **66.50** | **70.20** | **63.70** | **33.90** | **37.00** | **63.72** |

How does LinearRAG perform compared to state-of-the-art GraphRAG methods in terms of generation performance? **Q2** (Efficiency Analysis): How cost-efficient and time-efficient is LinearRAG relative to existing GraphRAG approaches? **Q3** (Ablation Study): What contribution does each component of LinearRAG make to the overall performance? ( Note that for a comprehensive evaluation of LinearRAG, additional experiments on retrieval quality, parameter sensitivity, backbone analysis, large-scale efficiency analysis and case studies are presented in Appendix E.)

## 4.1 EXPERIMENTAL SETTING

**Datasets.** We first evaluate the effectiveness of LinearRAG on three widely-used multi-hop QA datasets and one domain-specific dataset, including HotpotQA (Yang et al., 2018), 2WikiMulti-HopQA (2Wiki) (Ho et al., 2020), MuSiQue (Trivedi et al., 2022), and the Medical dataset from GraphRAG-Bench (Xiang et al., 2025). We follow the same evaluation method as HippoRAG, using the same corpus for retrieval and choosing 1,000 questions from each validation set. This setup allows for a fair comparison between different methods. We also test our approach on the domain-specific Medical dataset from GraphRAG-Bench (Xiang et al., 2025), showing that LinearRAG improves both generation results and retrieval quality.

**Baselines.** We categorize all the baselines into three groups: (i) Zero-shot LLM Inference: We evaluate several foundational models including LLaMA3 (8B) and LLaMA3 (13B) (Dubey et al., 2024), as well as GPT-3.5-turbo and GPT-4o-mini (OpenAI, 2023). (ii) We deploy Vanilla RAG methodologies across multiple retrieval configurations (retrieving 1, 3, or 5 top passages), combining semantic-based document retrieval with chain-of-thought reasoning prompts to guide the LLM's generation process. Additionally, we compare against advanced RAG methods, including RerankRAG, which incorporates a reranking model to refine retrieval results, and IRCOT (Trivedi et al., 2023) for iterative retrieval-reasoning. (iii) State-of-the-art GraphRAG Systems: We compare against leading GraphRAG implementations including KGP (Wang et al., 2024), G-retriever (He et al., 2024), RAPTOR (Sarthi et al., 2024), $E^2$GraphRAG (Zhao et al., 2025), LightRAG (Guo et al., 2024), HippoRAG (Gutiérrez et al., 2024), GFM-RAG (Luo et al., 2025a), and HippoRAG2 (Gutiérrez et al., 2025). Among these, RAPTOR organizes the corpus into a hierarchical tree graph; G-Retriever,

LightRAG, HippoRAG, GFM-RAG, and HippoRAG2 extract triples from passages to build structured graphs; while E$^2$GraphRAG integrates both strategies.

**Evaluation Metrics.** We evaluate our method using four metrics across two categories. For end-to-end QA performance, following existing work (Wang et al., 2025), we use: 1) Contain-Match Accuracy (Contain-Acc.), which checks if the correct answer appears in the generated response, and 2) GPT-Evaluation Accuracy (GPT-ACC.), an LLM-based metric that assesses whether the predicted answer matches the ground truth. For the Medical dataset, since golden answers consist of lengthy descriptive statements, we only evaluate using GPT-ACC. For retrieval quality assessment, we adopt metrics from GraphRAG-Bench (Xiang et al., 2025): 1) *Context Relevance*, which measuring semantic alignment between questions and retrieved passages, and 2) *Evidence Recall*, which evaluating whether the retrieved contents contain all necessary information for the correct answer.

**Implementations.** For consistency in implementation, all algorithms use the same embedding model, *i.e.*, all-mpnet-base-v2 (Xiao et al., 2023)). For the RerankRAG baseline, we employ the bge-reranker-large model for reranking. We set $k = 5$ for top $k$ retrieval in all methods. All RAG approaches employ the same LLM (GPT-4o-mini) for both generation and evaluation tasks. All experiments are conducted on the hardware configuration detailed in Appendix C.

## 4.2 GENERATION ACCURACY (Q1)

To address Q1, we conduct a comprehensive evaluation of generation performance by comparing various baseline methods with LinearRAG across four benchmark datasets. The detailed experimental results are presented in Table 1. Based on our analysis, we derive the following key observations.

**Obs. 1. RAG significantly enhances zero-shot LLM performance.** Direct prompting of LLMs without external knowledge retrieval yields the poorest performance across all datasets. For instance, advanced models like GPT-4o-mini achieve only 15.80% GPT-based accuracy on the MuSiQue dataset when operating without retrieval augmentation. However, integrating relevant corpus content into prompts substantially improves performance, as Vanilla RAG with top-5 retrieved contexts increases accuracy to 29.60% on the same dataset. Such significant performance enhancement demonstrates the essential nature of RAG mechanisms for information-intensive applications.

**Obs. 2. GraphRAG methods provide crucial context missed by RAG methods for complex reasoning.** While larger $k$ values improve accuracy, even for advanced RAG methods like RerankRAG and IRCOT that incorporate reranking and iterative retrieval modules, the gains diminish at higher values, revealing vanilla RAG's core limitation in multi-hop reasoning: it tends to overly focus on searching for mentioned entities or keywords within documents, thereby missing logic-related documents essential for the complete reasoning chain. By contrast, methods that model structural dependency in the retrieval consistently perform better. Among them, HippoRAG 2 achieves the best results across most datasets, attaining 62.90% Contain-based accuracy on HotpotQA and 31.00% on MuSiQue datasets.

**Obs. 3. LinearRAG demonstrates superior performance compared to GraphRAG methods, which suffer from sensitivity to constructed graph quality.** Typical GraphRAG methods address semantic misalignment by explicitly structuring knowledge into graphs, but their effectiveness relies heavily on relation extraction. Instead, LinearRAG gets rid of ineffective graph construction and LinearRAG outperforms all baselines by a significant margin across all datasets, empirically. Notably, LinearRAG achieves 63.70% GPT-based accuracy on 2Wiki, around 3.80% absolute improvement over the second best baseline, and boosts Contain-based accuracy to 70.20%.

## 4.3 EFFICIENCY ANALYSIS (Q2)

To better understand the associated efficiency and cost implications, we conduct a dedicated analysis on prompt statistics across different GraphRAG models during the indexing and retrieval stages by comparing their token usage and running time on 2WikiMultiHopQA dataset. The results are presented in Table 2, and we summarize our key observations as follows:

**Obs. 4. Complicated graph construction inevitably brings significant time costs during both the indexing and retrieval stages.** Models such as G-Retriever and LightRAG employ intricate schema definitions, resulting in notably poor efficiency. For instance, LightRAG requires 4,933.22

Table 2: **Efficiency and performance comparison of different GraphRAG methods.** Notably, Accuracy represents the average of Contain-Acc. and GPT-Acc. metrics. Prompt tokens represent input to LLM, and completion tokens represent LLM output. Best results are highlighted in **bold**, and second-best results are underlined.

| Method | Time (s) | | Token Consumption ($\times 10^6$) | | Accuracy |
|---|---|---|---|---|---|
| | Indexing | Retrieval (Avg.) | Prompt | Completion | |
| G-retriever | 2745.94 | 11.487 | 6.05 | 2.26 | 36.85 |
| RAPTOR | 1323.57 | 0.062 | 0.81 | 0.03 | 46.10 |
| E$^2$GraphRAG | 534.60 | **0.053** | 0.78 | 0.08 | 46.20 |
| LightRAG | 4933.22 | 10.963 | 35.52 | 51.16 | 47.10 |
| HippoRAG | 936.00 | 1.461 | 3.05 | 0.98 | 63.00 |
| GFM-RAG | 1202.77 | 1.211 | 3.05 | 0.98 | 63.20 |
| HippoRAG2 | 1147.01 | 1.694 | 4.98 | 1.22 | 58.85 |
| **LinearRAG (Ours)** | **249.78** | 0.093 | **0** | **0** | **66.95** |

seconds for indexing and 10.963 seconds for retrieval. These models define entities and keywords within the graph and perform both high-level and low-level retrieval. Consequently, LLMs are overburdened with processing the large volume of content retrieved from these two information sources, which results in higher fees and latency.

**Obs. 5. Reducing LLM token costs does not necessarily have a negative impact on performance.** Complicated prompt construction and completion in G-Retriever and LightRAG fail to deliver consistent performance improvements. For instance, HippoRAG2 uses only 3.05M tokens during prompt construction and 0.98M tokens in the completion stage, while HippoRAG2 consumes 4.98M tokens for prompt construction and 1.22M tokens in completion. Despite these lower token costs, both models outperform G-Retriever and LightRAG.

**Obs. 6. LinearRAG delivers the strongest overall efficiency.** Considering the whole pipeline, LinearRAG is the fastest and introduces zero token usage. While E$^2$GraphRAG and RAPTOR are more time-efficient than LinearRAG at the retrieval stage, it comes at the cost of significantly reduced model performance, as it retrieves only directly related documents with the query without considering the structural dependencies in the reasoning chain. In comparison, the lightweight pipeline of LinearRAG avoids LLM calls for both indexing and retrieval, which minimizes latency and eliminates token costs. For deployments that demand strong performance along with speed, scalability, and cost control, LinearRAG is the most practical choice.

## 4.4 ABLATION STUDY (Q3)

**Effect of core components in LinearRAG.** We conduct systematic ablation studies on the core components of LinearRAG across four datasets. We examine two key modules: ❶ **Relevant Entity Activation via Semantic Bridging**, *i.e., w/o Entity Activation*, which directly uses the initial entities extracted from the query as activated entities, bypassing relevant entity activation via semantic bridging, and moving directly to the passage retrieval stage. ❷ **Passage Retrieval via Global Importance Aggregation**, *i.e., w/o Global Importance Aggregation*, which skips the global importance and di-

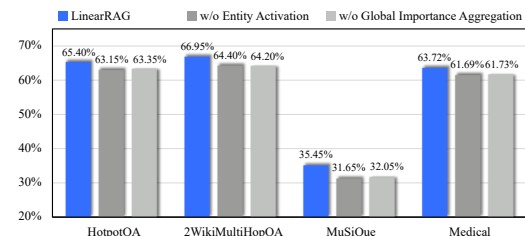

Figure 4: **Ablation study on key modules of LinearRAG under four different datasets.** The y-axis represents the average of GPT-Acc. and Contain-Acc.

rectly uses the scores computed by the entity activation stage to retrieve documents from the corpus, without considering global importance relationships. The experimental results are shown in Figure 4, and we have the following findings.

**Obs. 7. Each module of LinearRAG is critical for optimal performance.** The performance gains are primarily attributed to two complementary stages. In the first Stage, LinearRAG identifies contextually relevant entities through semantic similarity propagation, uncovering the hidden logical

relationships in multi-hop questions. In the second Stage, the model uses personalized PageRank algorithms on the activated subgraph to evaluate document importance from a global perspective. These two modules serve different but complementary purposes, allowing LinearRAG to achieve both effective and efficient performance.

**Impact of Pruning Strategies in LinearRAG.** As mentioned in Sec. 3.2.1, LinearRAG employs a threshold $\delta$ in the entity activation stage to prune entity nodes with weak relevance. In this part, we investigate this strategy by comparing it with two variants: 1) *w/o pruning*, which retains all activated entities and uses a fixed number of iterations (e.g., 3 iterations), and 2) *w. dynamic threshold*, which employs a threshold that decays with the number of iterations to examine the importance of using a fixed threshold. Specifically, we explore two decay schemes: geometric decay ($\times 0.5$) and linear decay ($-0.1$).

Table 3: **Performance (%) and efficiency comparison of different pruning strategies on 2Wiki dataset.** "Acc" means the average scores of GPT-Acc. and Contain-Acc.

| Method | Acc. | Avg. Retrieval Time (s) |
|---|---|---|
| **LinearRAG** | **66.95** | **0.093** |
| w/o pruning | 64.50 | 0.186 |
| w. dynamic threshold ($\times 0.5$) | 65.15 | 0.123 |
| w. dynamic threshold ($-0.1$) | 65.35 | 0.095 |

**Obs. 8. The fixed threshold pruning strategy in LinearRAG consistently outperforms both no pruning and dynamic threshold approaches.** As shown in Table 3, LinearRAG with fixed threshold $\delta$ achieves the best overall performance. The *w/o pruning* variant suffers significant performance drops, indicating that retaining all weakly-relevant entities introduces noise and hinders reasoning. The dynamic threshold strategies also underperform compared to the fixed threshold, demonstrating that an explicit distance-decaying threshold is unnecessary and potentially harmful because it could prematurely discard valid long-range bridges that are crucial for multi-hop reasoning or introduce more noise, while the selection of the decay factor itself is difficult and may negatively affect the model's generalization ability across different datasets.

**Impact of using different LLM as evaluators.** To address the concern regarding potential evaluation bias from relying on a single LLM evaluator, we conduct additional experiments using three different state-of-the-art LLMs: GPT-4o-mini (OpenAI, 2023), DeepSeek-Chat (Bi et al., 2024), and Claude-Haiku-4.5 and compare LinearRAG against the most advanced baselines (HippoRAG2, GFMRAG, and RAPTOR) across four benchmarks.

Table 4: **Generation accuracy (%) across different LLM evaluators.** Notably, the values represent GPT-Acc.

| Method | LLM Evaluator | HotpotQA | 2Wiki | MuSiQue | Medical |
|---|---|---|---|---|---|
| HippoRAG2 | gpt-4o-mini | 64.30 | 55.00 | 35.00 | 60.77 |
| | deepseek-chat | 69.70 | 62.30 | 38.80 | 64.40 |
| | claude-haiku-4.5 | 63.60 | 58.50 | 35.60 | 62.12 |
| GFMRAG | gpt-4o-mini | 65.60 | 59.60 | 34.60 | 56.07 |
| | deepseek-chat | 70.70 | 68.10 | 36.50 | 60.58 |
| | claude-haiku-4.5 | 63.60 | 64.80 | 34.30 | 56.84 |
| RAPTOR | gpt-4o-mini | 58.30 | 42.10 | 27.40 | 55.75 |
| | deepseek-chat | 62.90 | 48.60 | 30.40 | 58.39 |
| | claude-haiku-4.5 | 57.50 | 44.90 | 27.00 | 56.21 |
| **LinearRAG** | gpt-4o-mini | **66.50** | **63.70** | **37.00** | **63.72** |
| | deepseek-chat | **72.60** | **70.20** | **39.90** | **66.54** |
| | claude-haiku-4.5 | **65.90** | **66.20** | **36.90** | **64.11** |

**Obs. 9. LinearRAG consistently outperforms all baseline methods across all three LLM evaluators, demonstrating robustness against evaluation bias.** As shown in Table 4, LinearRAG maintains its performance advantage regardless of which LLM is used as the evaluator. While different evaluators exhibit varying scoring patterns (e.g., DeepSeek-Chat tends to assign higher scores overall compared to Claude-Haiku-4.5), the relative ranking remains consistent.

## 5 CONCLUSION

In this work, we introduce LinearRAG, a novel GraphRAG framework that simplifies graph construction by replacing costly and error-prone relation extraction with lightweight entity extraction. It creates a hierarchical graph encompassing entities, sentences, and passages. Upon this, LinearRAG features a two-stage retrieval mechanism that advances the precision-recall Pareto frontier by jointly leveraging both local and global structural information. Extensive experiments demonstrate that LinearRAG consistently surpasses state-of-the-art baselines in retrieval precision, generation accuracy, and scalability, offering a robust and efficient solution for handling complex queries.

## ETHICS STATEMENT

The first three datasets used in the experiments including HotpotQA, 2WikiMultiHopQA and MuSiQue are widely used datasets. The medical benchmark dataset is built from public resources. Our research strictly adheres to the ICLR Code of Ethics, particularly regarding data privacy, transparency, and responsible computing practices. And there is no participant involved.

## REPRODUCIBILITY STATEMENT

We provide detailed information required to reproduce the main experimental results of the paper, including data splits and hyperparameter configurations. Additionally, for each experiment, we outline the required computational resources, such as memory usage and execution time. All baseline models are sourced from public repositories. Our code and dataset are made available. Besides that, the code and implementation details of our model are available at `https://github.com/DEEP-PolyU/LinearRAG`.

## ACKNOWLEDGEMENT

The work described in this paper was fully supported by a grant from the Innovation and Technology Commission of the Hong Kong Special Administrative Region, China (Project No. GHP/391/22).

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

# A    DATASETS

Our experimental evaluation is conducted on four datasets: three established multi-hop benchmark datasets for multi-hop question answering—HotpotQA (Yang et al., 2018), MuSiQue (Trivedi et al., 2022), and 2WikiMultiHopQA (2Wiki) (Ho et al., 2020)—and one domain-specific dataset. Below, we provide a concise overview of each dataset's key characteristics.

**(i) HotpotQA (Yang et al., 2018):** A comprehensive benchmark comprising 97k question-answer instances designed to evaluate multi-hop reasoning capabilities. Each question requires models to synthesize information from multiple documents, with up to 2 gold-standard supporting passages provided alongside numerous irrelevant documents. This structure challenges systems to perform effective cross-document inference and evidence selection.

**(ii) 2WikiMultiHopQA (2Wiki) (Ho et al., 2020):** A multi-hop reasoning benchmark containing 192k questions that necessitate information integration across multiple Wikipedia articles. Each instance requires evidence synthesis from either 2 or 4 specific articles, testing models' ability to perform structured cross-document reasoning and maintain coherent information flow.

**(iii) MuSiQue (Trivedi et al., 2022):** A sophisticated multi-hop QA benchmark featuring 25k question-answer pairs that demand 2-4 sequential reasoning steps. Each question requires coherent multi-step logical inference across multiple documents, challenging systems to execute structured reasoning chains while preserving contextual consistency throughout the inference process.

**(iv) Medical:** A specialized subset derived from GraphRAG-Bench (Xiang et al., 2025), constructed from structured clinical data sourced from the National Comprehensive Cancer Network (NCCN) guidelines. These guidelines provide standardized treatment protocols, drug interaction hierarchies, and diagnostic criteria. The dataset encompasses four tasks of increasing complexity: fact retrieval, complex reasoning, contextual summarization, and creative generation, totaling 4,076 questions across all difficulty levels.

# B    BASELINE DETAILS

In our experiments, we compare our method against several RAG and GraphRAG approaches.

**(i) RerankRAG** incorporates a reranking model to refine retrieval results.

**(ii) IRCOT** (Trivedi et al., 2023) employs an iterative retrieval-reasoning framework that interleaves retrieval and chain-of-thought reasoning for multi-step question answering.

**(iii) KGP** (Wang et al., 2024) builds a knowledge graph over multiple passages with LLMs; at retrieval stage, it introduces an LLM-driven graph traversal agent to navigate the graph and progressively collect supporting passage.

**(iv) G-Retriever** (He et al., 2024) combines graph neural networks with LLMs by formulating subgraph retrieval as a Prize-Collecting Steiner Tree optimization problem, enabling effective conversational question answering on textual graphs while mitigating hallucination and enhancing scalability.

**(v) RAPTOR** (Sarthi et al., 2024) develops a hierarchical tree by applying clustering algorithms and abstractive summarization techniques, facilitating representation at multiple semantic granularities.

**(vi) $E^2$GraphRAG** (Zhao et al., 2025) uses spaCy to extract entities and LLMs to summarize passage groups into a hierarchical tree with encoded nodes at indexing stage; at retrieval stage, it hits relevant entities in $k$-hop neighborhood and collects associated passages for ranking, otherwise dense retrieval over the whole tree.

**(vii) LightRAG** (Guo et al., 2024) employs a two-tier framework that incorporates graph-based representations within textual indexing, merging fine-grained entity-relation mappings with coarse-grained thematic structures.

**(viii) HippoRAG** (Gutiérrez et al., 2024) is a training-free graph-enhanced retriever that uses the Personalized PageRank algorithm with query concepts as seeds for single-step or multi-hop retrieval across disparate documents.

**(ix) GFM-RAG** (Luo et al., 2025a) implements a GraphRAG paradigm by constructing graphs from documents and using a graph-enhanced retriever to retrieve relevant documents.

**(x) HippoRAG2** (Gutiérrez et al., 2025) extends HippoRAG with enhanced paragraph integration and contextualization. Optimizes seed node selection and PageRank reset probabilities while maintaining factual memory capabilities and improving associative memory performance. improving associative memory performance.

## C  MACHINE CONFIGURATION

All experiments in this study were conducted on the hardware configuration detailed in Table 5.

Table 5: Detailed machine configuration used in our experiments.

| Component | Specification |
|---|---|
| GPU | NVIDIA GeForce RTX 4090 D (24GB VRAM) |
| CPU | Intel(R) Xeon(R) Gold 6426Y |

## D  EFFICIENCY ANALYSIS OF GRAPH CONSTRUCTION: ALL-STAGE LINEAR SCALABILITY

We present the efficiency analysis of LinearRAG and demonstrate that it achieves linear scalability in both runtime and memory consumption during graph construction and retrieval, thereby ensuring all-stage linear scalability. Detailed analysis is provided below.

**Graph construction stage** involves lightweight operations such as sentence segmentation and named entity recognition over the corpus. The resulting **computational complexity** is $O(|\mathcal{P}| \cdot T)$, where $T$ denotes the average length of each passage. Notably, this stage incurs no LLM token consumption. Regarding memory usage, the first data structures to be stored are the embeddings of passages, nodes, and sentences, all of which scale linearly with the corpus size, i.e., $O(|\mathcal{P}| \cdot T)$. The second set of data structures includes two adjacency matrices, $M$ and $C$, stored in sparse format by leveraging the inherent sparsity (each sentence contains at most $\sim 4$ entities, and each passage at most $\sim 10$ entities). Thus, memory consumption is $O(|\mathcal{P}| + |\mathcal{S}|)$, which simplifies to $O(|\mathcal{P}|)$ since the number of sentences is proportional to the number of passages. Therefore, the **overall memory complexity** is $O(|\mathcal{P}| \cdot T + |\mathcal{P}|) = O(|\mathcal{P}| \cdot T)$, i.e., linear with respect to corpus size.

**Retrieval stage** involves similarity computations, semantic propagation using SpMM (sparse matrix multiplication), and Personalized PageRank (PPR) iterations. Each propagation step requires $O(\text{nnz})$ time, where nnz is the number of non-zero entries in the sparse matrices ($O(|\mathcal{S}|)$), and PPR on the bipartite graph is computable in linear time with respect to the number of edges ($O(|\mathcal{P}|)$). Overall, the **computational complexity** is $O(n \times |\mathcal{S}|)$, where $n$ is the number iterations of equation 5. Memory consumption primarily arises from loading sparse graph structures, activation vectors (of size $O(|\mathcal{E}| + |\mathcal{S}| + |\mathcal{P}|)$), and temporary query-specific computations.

**Acceleration with parallel computation.** The above analysis confirms the linear scalability of LinearRAG in both graph construction and retrieval. Additionally, due to the vectorized formulation of the graph structures, all data manipulations can be efficiently accelerated through parallel computation, further boosting performance in both stages. This is empirically verified in our efficiency experiments in Section 4.3, Appendix E.4 and Appendix E.5.

## E  ADDITIONAL EXPERIMENTS

### E.1  RETRIEVAL QUALITY EVALUATION (Q4)

To provide a more comprehensive assessment beyond generation performance, we adopt the settings from GraphRAG-Bench and utilize four tasks of increasing difficulty: Fact Retrieval, Complex Reasoning, Contextual Summarization, and Creative Generation. The comparison is conducted on the

Table 6: **Retrieval quality evaluation results (%) of different baselines across four different question categories.** The table compares recall and relevance metrics for fact retrieval, complex reasoning, contextual understanding, and creative generation tasks. The best results are highlighted in **bold**, and second-best results are underlined.

| Method | Fact Retrieval | | Complex Reasoning | | Contextual | | Creative Generation | |
|---|---|---|---|---|---|---|---|---|
| | Recall | Relevance | Recall | Relevance | Recall | Relevance | Recall | Relevance |
| Vanilla RAG (Top-5) | 86.24 | 63.71 | 84.97 | **84.11** | 84.14 | **89.94** | 44.88 | 58.73 |
| RAPTOR | 85.40 | 69.38 | **89.70** | 53.20 | 88.86 | 58.73 | 72.70 | 52.71 |
| E$^2$GraphRAG | 87.84 | 69.74 | 87.08 | 62.67 | **89.17** | 71.63 | 60.26 | 35.84 |
| LightRAG | 80.32 | 41.27 | 82.91 | 42.79 | 85.71 | 43.11 | 81.34 | 45.17 |
| HippoRAG | 87.25 | 52.44 | 83.80 | 42.19 | 83.46 | 49.13 | 81.66 | 45.03 |
| GFM-RAG | **90.08** | 57.90 | 85.03 | 33.06 | 78.62 | 40.14 | 83.51 | 22.87 |
| **LinearRAG (Ours)** | 88.86 | **86.09** | 87.03 | 81.58 | 89.13 | 87.89 | **89.08** | **72.74** |

Medical dataset, same as GraphRAG-Bench. The results comparing LinearRAG with representative GraphRAG methods are presented in Table 6 and we have the following observations.

**Obs. 10. GraphRAG models demonstrate significant improvements in recall metrics compared to RAG baselines, particularly in creative generation tasks, but fall behind in relevance metrics.** We infer that the reasons are twofold. On the one hand, for simpler tasks such as fact retrieval, models are not required to analyze reasoning chains to recall all the necessary supporting facts. Direct semantic retrieval is enough, reducing the risk of retrieving irrelevant or redundant information during graph traversal. On the other hand, for more complex tasks such as creative generation, the model needs continuous documents with strong logical dependencies to produce satisfactory outputs. RAG struggles to capture such complex relationships due to underlying graph structure of gold answers. Specifically, GFM-RAG improves recall from 44.88% to 83.51% in creative generation, but relevance drops from 58.73% to 22.87%.

**Obs. 11. LinearRAG demonstrates tremendous performance improvements over GraphRAG models in relevance metrics, owning the advantages of both high recall and strong relevance.** It is challenging to improve recall and relevance simultaneously because increasing recall often introduces more irrelevant documents, lowering relevance, while increasing relevance usually misses relevant results and lowering recall. LinearRAG achieves the best performance across four tasks in recall while achieves better than RAG in relevance metrics in most cases. We infer that the reason lies in our construction of a high-quality graph with minimal redundant connections, ensuring that irrelevant documents are not retrieved. Specifically, in complex reasoning tasks, LinearRAG achieves 87.03% recall and 81.58% relevance, significantly outperforming GFM-RAG's 85.03% recall but only 33.06% relevance, demonstrating LinearRAG's ability to maintain high precision while preserving comprehensive information coverage.

### E.2 HYPER-PARAMETER SENSITIVITY (Q5)

We conduct ablation studies to investigate the impact of key hyperparameters in LinearRAG. All experiments are evaluated using the average scores of GPT-Acc. and Contain-Acc. on 2WikiMultiHopQA.

**Obs. 12. Impact of threshold $\delta$.** The threshold $\delta$ determines whether an entity's score is retained during dynamic pruning and whether to continue expansion. As shown in Figure 5a, when $\delta$ is too small, it introduces excessive noise and reduces efficiency in the retrieval stage. However, when $\delta$ is too large, it prevents

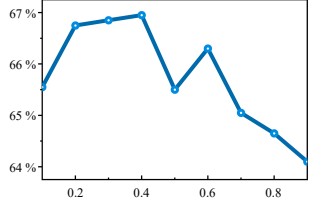

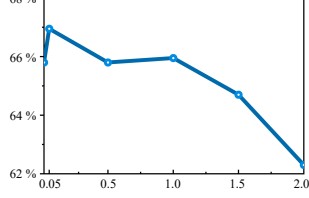

(a) The impact of threshold $\delta$.     (b) The impact of coefficient $\lambda$.

Figure 5: **Parameter analysis of LinearRAG performance in the 2WikiMultiHopQA dataset.** (a) shows the dependency of LinearRAG performance on threshold $\delta$ in dynamic pruning. (b) examines the effect of trade-off coefficient $\lambda$.

Table 7: **Comparison of different sentence embedding models used in LinearRAG.** The best results are highlighted in **bold**.

| Model | HotpotQA | | 2Wiki | | MuSiQue | | Medical |
|---|---|---|---|---|---|---|---|
| | Contain-Acc. | GPT-Acc. | Contain-Acc. | GPT-Acc. | Contain-Acc. | GPT-Acc. | GPT-Acc. |
| all-mpnet-base-v2 | 64.30 | 66.50 | **70.20** | **63.70** | **33.90** | **37.00** | 63.72 |
| all-MiniLM-L6-v2 | 64.20 | 64.90 | 69.50 | 62.60 | 32.80 | 36.90 | 62.32 |
| bge-large-en-v1.5 | 66.20 | 67.60 | 69.80 | 63.90 | 31.80 | 35.20 | 64.45 |
| e5-large-v2 | **66.80** | **68.30** | 69.90 | 63.10 | 31.70 | 36.50 | **65.42** |

Table 8: **Indexing efficiency comparison of GraphRAG methods on large-scale ATLAS-Wiki corpus.** Results are evaluated on ATLAS-Wiki subsets of 5M and 10M tokens. Notaly, prompt tokens represent input to LLM, and completion tokens represent LLM output.

| Dataset | Method | Indexing Time (s) | Token Consumption ($\times 10^6$) | |
|---|---|---|---|---|
| | | | Prompt | Completion |
| 5M | RAPTOR | 18033.75 | 7.43 | 0.96 |
| | HippoRAG | 6032.46 | 13.94 | 3.68 |
| | **LinearRAG (Ours)** | **1409.95** | **0** | **0** |
| 10M | RAPTOR | 46430.96 | 16.62 | 2.82 |
| | HippoRAG | 13815 | 28.04 | 7.53 |
| | **LinearRAG (Ours)** | **3084.38** | **0** | **0** |

the incorporation of more relevant entities, limiting the model's ability to capture comprehensive contextual information. Therefore, selecting an appropriate value for $\delta$ is crucial for balancing retrieval quality and computational efficiency. And, we set $\delta = 4$ based on our experimental setting.

**Obs. 13. Impact of trade-off coefficient** $\lambda$**.** The parameter $\lambda$ controls the balance between DPR-based passage similarity and entity-level information during passage initial scoring. As observed in Figure 5b, optimal performance is achieved when $\lambda$ takes a relatively small value (e.g., 0.05), ndicating that entity information serves as the primary component while DPR similarity score acts as an auxiliary enhancement.

### E.3 EFFECTIVENESS OF DIFFERENT SENTENCE EMBEDDINGS (Q6)

In this section, we study the effectiveness of different sentence embeddings in the LinearRAG. We compare the all-mpnet-base-v2 (Song et al., 2020), bge-large-en (Xiao et al., 2023), all-MiniLM-L6-v2 (Wang et al., 2020) and e5-large-v2 (Wang et al., 2022). We download the official pre-trained model from the Huggingface2.

**Obs. 14.** Based on the findings presented in Table 7, we observe that performance variations across different sentence embedding models remain relatively modest, with all-mpnet-base-v2 demonstrating superior results on the majority of datasets. This suggests that LinearRAG exhibits robustness to the selection of sentence embedding models. Consequently, we adopt all-mpnet-v2 as our default embedding models in all experiments due to its computational efficiency.

### E.4 LARGE-SCALE EFFICIENCY ANALYSIS (Q7)

To evaluate the scalability and effectiveness of LinearRAG on large corpora, we conduct comprehensive efficiency analysis using the ATLAS-Wiki dataset (Bai et al., 2025). We create two subsets containing 5M and 10M tokens respectively for indexing phase evaluation, as shown in Table 8.

**Obs. 15.** The results demonstrate that LinearRAG achieves significant efficiency advantages over typical GraphRAG methods. Specifically, LinearRAG eliminates token consumption entirely (0 prompt and completion tokens) while achieving 12.8× and 15.1× speedup compared to RAPTOR on 5M and 10M datasets respectively. This zero-cost indexing approach makes LinearRAG particularly suitable for large-scale enterprise deployments without API dependencies.

### E.5 RETRIEVAL EFFICIENCY ACROSS GRAPH SIZES AND DOMAINS (Q8)

In this section, we select four benchmarks of varying scales: Medical, 2WikiMultiHopQA, Hot-potQA, and Novel (Xiang et al., 2025), to evaluate the average retrieval latency across different graph sizes and domains. As shown in Table 9, the corpus sizes range from 202K to 1.06M tokens, where Medical is from the medical domain, Novel is from literary texts, and 2WikiMultiHopQA and HotpotQA are from Wikidata knowledge.

**Obs. 16.** We observe that retrieval latency increases with graph size, ranging from 0.081s for the smallest dataset (Medical, 202K nodes) to 0.107s for the largest (HotpotQA, 1.06M nodes). Notably, the retrieval process remains highly efficient across different sizes and domains, taking approximately 0.1s even on the largest graph, which demonstrates the practical scalability of LinearRAG.

Table 9: Graph size and retrieval latency statistics across datasets.

| Dataset | Graph Size | Avg. Retrieval Time (s) |
|---------|-----------|-------------------------|
| Medical | 202,126 | 0.081 |
| 2Wiki | 517,370 | 0.093 |
| Novel | 1,027,013 | 0.102 |
| HotpotQA | 1,062,480 | 0.107 |

### E.6 CASE STUDY (Q9)

Table 10: **Case study of LinearRAG.** LinearRAG successfully captures implicit relationships without requiring explicit relation extraction, enabling correct multi-hop reasoning.

| Question | "What nationality is Beatrice I, Countess Of Burgundy's husband?" |
|----------|------------------------------------------------------------------|
| **Ground Truth** | Germany |
| **Support Context** | ["Beatrice I, Countess of Burgundy" → "spouse" → "Frederick Barbarossa"]
["Frederick Barbarossa" → "country of citizenship" → "Germany"] |
| **HippoRAG2** | *Retrieved context:*
1) ✓ "William Duncan (actor)": ...frederick barbarossa was elected king of germany at frankfurt on 4 march 1152...
2) ✗ "Dulce of Aragon": "Dulce of Aragon (or of Barcelona; ; 1160 Ž2013 1 September...
3) ✗ "Gordon Flemyng": Gordon William Flemyng( 7 March 1934 Ž2013 12 July 1995) was a...
4) ✗ "Marian Hillar": Marian Hillar is an American philosopher, theologian, linguis...
5) ✗ "Semi-Tough": Semi-Tough is a 1977 American comedy film directed by Michael...
*Prediction:*
✗ French. |
| **LinearRAG** | *Activated Entities:*
Tter 0-Entity: Beatrice I
Tter 0-Sentence: Beatrice I (1143 Ž2013 15 November 1184) was Countess of Burgundy from 1148 until her death, and was also Holy Roman Empress by marriage to Frederick Barbarossa.
Tter 1-Entity: Frederick Barbarossa
Tter 1-Sentence: Frederick Barbarossa was elected King of Germany at Frankfurt on 4 March 1152 and crowned in Aachen on 9 March 1152.
Tter 2-Entity: Germany
*Retrieved context:*
1) ✓ "William Duncan (actor)": ...he was elected king of germany at frankfurt on 4 march 1152...
2) ✓ "Beatrice I, Countess of Burgundy": ...holy roman empress by marriage to frederick barbarossa...
3) ✗ "Richilde, Countess of Hainaut": Richilde, Countess of Mons and Hainaut( Ž2013 15 March 1086)...
4) ✗ "Sophie of Pomerania, Duchess of Pomerania": Sophia of Pomerania- Stolp( 1435 Ž2013 24...
5) ✗ ""DŎ0e1ire Drechlethan": "DŎ0e1ire DrechlethanDŎ0e1ire of the Broad Faceïs a...
*Prediction:*
✓ Germany. |

To clearly contrast typical GraphRAG baselines with our LinearRAG framework, we include a detailed case analysis in Table 10, comparing results from the strong baseline HippoRAG2 and our model on a multi-hop question from the 2WikiMultihopQA dataset.

**Obs. 18.** The case demonstrates that HippoRAG2 fails to retrieve relevant evidence due to its dependence on explicitly pre-extracted relations (such as husband) that are missing in this context. In comparison, our LinearRAG method effectively captures implicit relationships through entity activation and contextual chaining, without dependence on pre-extracted relational tuples.

## F  RELATED WORK

Large language models are prone to hallucination (Fang et al., 2024; Jiang et al., 2025; Fang et al., 2025; Zheng et al., 2025; Hong et al., 2024; Yuan et al., 2025; Zhong et al., 2024), while RAG is a promising solution by grounding the reasoning process on contextual evidence from knowledge bases (Zhang et al., 2025a; Zhou et al., 2025c; Zhang et al., 2025b; Yang et al., 2026; Luo et al., 2025b; Xia et al., 2025; Luo et al., 2024). However, real-world knowledge is often distributed across documents, organizing them effectively for answering complex questions has always been a challenging yet promising research topic in RAG. In the following, we discuss two major lines (❶, ❷) of graphRAG research that closely relate to our work, which construct explicit graphs for organizing external knowledge sources. Then discuss reasoning-enhanced RAG (❸), which leverages LLM's inherent reasoning ability for answering complex questions without explicitly structuring corpora.

❶ **Clustering-based hierarchy construction.** One line of research employs clustering-based community detection, a bottom-up method that applies algorithms like Louvain or Leiden to identify densely connected clusters of entities within the initial graph (Edge et al., 2024; Sarthi et al., 2024; Gutiérrez et al., 2024). By grouping related entities, it creates a hierarchical structure that abstracts passages into topic-based communities, reducing redundancy and offering a broader, macro-level view of the data Zhou et al. (2024). However, as an unsupervised technique, it is vulnerable to error propagation Zhou et al. (2025b), where inaccuracies in entity relationships are amplified at higher levels of abstraction. Moreover, applying these clustering algorithms to large-scale graphs presents significant scalability issues, making real-time applications infeasible in practice.

❷ **Relation-extraction-based knowledge graph construction.** This line of research (Gutiérrez et al., 2024; Guo et al., 2024; Xiao et al., 2025c) adopts the idea of knowledge graphs to organize knowledge across different passages. The basic idea is to extract triples from each text chunk (passage) as an atomic summarization of the knowledge within the passage. The triples are connected via entity alignment (Chen et al., 2025a; 2024), ultimately forming a unified knowledge graph, which directly carries structured knowledge and serves as the index of passages, where off-the-shelf graph reasoning algorithms (Shengyuan et al., 2024; Qu & Tang, 2019; Liu et al., 2023) can be applied to perform reasoning (Sun et al., 2024; LUO et al., 2024). However, due to context window limits, OpenIE on each passage is processed independently, so the generated triples may be inconsistent. Recent work (Liang et al., 2024; Sharma et al., 2024) mitigates this via top-down graph construction guided by a globally defined schema, but it relies on manual expert annotation to create and maintain the domain-specific schema, which is not only expensive, time-consuming, and does not generalize well across domains.

❸ **Reasoning-enhanced RAG.** The key to answering complex questions in RAG is to effectively identify multiple distributed documents that support multihop logical dependencies. While GraphRAG methods explicitly construct graphs with human priors to organize corpora for efficient structured retrieval, reasoning-enhanced RAG directly leverages the inherent reasoning ability of LLMs to decompose complex queries into simpler subquestions that can be addressed by dense retrieval. LogicRAG (Chen et al., 2025b) and LAG (Xiao et al., 2025b) formalize this idea by first decomposing a query into a set of subqueries, then constructing a directed acyclic graph based on the logical dependencies among them, and solving subproblems one by one by following the topological order. Similarly, Chain-of-Note (CoN) (Yu et al., 2024) enhances RAG by generating sequential notes that break down complex queries into intermediate reasoning steps, retrieving relevant documents for each step to build a coherent answer. Another approach, SelfRAG (Asai et al., 2024), integrates self-reflection into the RAG process, where the LLM iteratively evaluates and refines subqueries to ensure retrieved documents align with the logical flow of the reasoning process. These methods collectively highlight the potential of LLM-driven reasoning to improve retrieval efficiency and answer quality for complex, multihop queries.

## G  FREQUENTLY ASKED QUESTIONS (FAQS)

### G.1  HOW DOES LINEARRAG HANDLE MULTI-HOP TASKS?

Our two-stage design, which consists of local semantic bridging followed by global importance aggregation, is built to capture long-range dependencies without requiring explicit relation triples. The first stage functions as a progressive clue-following mechanism that performs iterative entity activa-

tion on the sentence–entity bipartite graph, exploring an n-hop subgraph to anchor relevant context passages. The second stage then scores each passage by evaluating its relevance to the activated entities as well as its semantic similarity to the query. PPR is then applied on the passage–entity bipartite graph to propagate importance from activated entities and rank passages globally. A concrete example illustrating how this mechanism works in practice is provided in Table 10.

## G.2 How does LinearRAG address the polysemy problem?

We acknowledge that polysemy is a common challenge in information retrieval systems, where different passages may be incorrectly connected through entities sharing the same name, such as "Apple" referring to both the fruit and the company. However, our design can largely alleviate this issue from the following aspects:

- By deferring relation understanding to the LLM during generation, our approach leverages the LLM's semantic understanding ability to distinguish conceptual differences between entities by analyzing the original context, effectively filtering noise information from irrelevant passages.

- LinearRAG employs sentence-level semantic bridging during entity activation, judging entity relevance through the context of sentences. Homonymous entities with different meanings are naturally distinguished by the semantic differences in their surrounding sentences.

- LinearRAG can naturally integrate self-reflection and query-rewrite strategies as post-refinement for the RAG process.

## G.3 Why adopts binary sentence-entity edge weights in LinearRAG?

Some may argue that using binary weights for sentence-entity edges is a simplification that fails to capture the varying semantic importance of different entities, potentially amplifying the influence of noisy or peripheral entities during propagation. We address this concern from three perspectives:

- Binary sentence-entity edges are adopted to preserve efficiency guarantees, specifically, fully LLM-free indexing and linear-time construction during graph building. Introducing richer precomputed weights, such as TF-IDF or LLM-based scores, would require additional computation and thus break these guarantees.

- Assigning fixed importance weights without knowing the incoming query risks down-weighting peripheral entities that may serve as critical intermediate anchors in a particular query. Binary weighting treats all mentioned entities equally and therefore preserves the possibility of activating less-frequent but query-relevant bridges, favoring recall especially on multi-hop benchmarks.

- A typical sentence contains only two to three entities, naturally limiting the fan-out of noisy entities and making binary weighting far less harmful. Additionally, dynamic pruning as described in Section 3.2.1 requires entities to exceed a relevance threshold to be carried forward, further filtering peripheral or noisy entities.

## G.4 How does LinearRAG handle queries with no extractable entity?

When no entities can be extracted from the query, the retrieval process does not halt. LinearRAG automatically falls back to standard similarity-based dense passage retrieval, ensuring the system remains robust even for entity-free or highly abstract questions. Furthermore, when extracted entities are ambiguous and cannot be disambiguated without the full query context, our design mitigates this issue through two mechanisms. During entity activation, we compute the query-sentence relevance distribution, which is used for semantic propagation, so ambiguously activated entities receive low support and do not effectively propagate relevance to their neighbors, while correctly activated entities dominate the propagation. In the global importance aggregation stage, as described in Section 3.2.2, passage importance scores are initialized with direct query-passage similarity. This additional signal incorporates the complete query context and further suppresses passages reached only through ambiguous or incorrect entities.

## H    THE USE OF LARGE LANGUAGE MODELS

We employ LLMs primarily for writing polish, including correcting spelling errors, fixing grammatical issues, and rewriting non-native expressions to improve clarity and fluency. And LLMs are used only for writing polishing of our manuscript and appendix. It does not generate research ideas, results, or claims. We ensure that all scientific contributions and implementations are original.

