# OpenReview forum: "LinearRAG: Linear Graph Retrieval Augmented Generation on Large-scale Corpora"
_ICLR.cc/2026/Conference — ICLR 2026 Poster_

### Official Review · Reviewer_df73 · 2025-10-29

**Soundness:** 3
**Presentation:** 3
**Contribution:** 3
**Rating:** 6
**Confidence:** 5

**Summary:**

This paper introduces LinearRAG, a GraphRAG framework that streamlines graph construction by employing lightweight entity extraction instead of complex relation extraction. It constructs a hierarchical graph comprising entities, sentences, and passages and utilizes a two-stage retrieval strategy that integrates both local and global structural information to enhance precision and recall.

**Strengths:**

S1: Addresses a timely and significant challenge relevant to both industry and academia.
S2: The proposed method is straightforward, achieving the fastest indexing speed alongside very low retrieval latency.
S3: Demonstrates competitive performance in token efficiency, content accuracy, and evaluation accuracy using large language models (LLMs) as judges.

**Weaknesses:**

W1: Some equations lack clarity and contain unexpected errors.
W2: The effectiveness of the dynamic pruning mechanism is not empirically evaluated.
W3: Since LLMs serve as the evaluators, it would be beneficial to incorporate multiple state-of-the-art LLMs from diverse vendors to mitigate potential evaluation bias.

**Questions:**

1. Entity activation is a crucial initial step in LinearRAG. However, the authors do not clarify how the framework handles queries from which no entities can be extracted. Does the retrieval process halt in the absence of seed entities? Additionally, if extracted entities are ambiguous and cannot be disambiguated without the query’s full context, could this lead to activation of incorrect entities and degraded retrieval performance? An explanation addressing these scenarios would strengthen the paper.

2. In Equation 5, the matrix M is not defined. Moreover, since σ_q is a ∣S∣×1 vector (per Equation 4), its transpose σ_q^T is 1×∣S∣, which mismatches the dimension ∣V_e∣ of α_q^(t-1) (from Equation 3). A more detailed explanation of this equation and its dimensions is necessary.

3. LinearRAG’s dynamic pruning currently applies a fixed threshold across all expansions. Given that dynamic pruning aims to produce high-quality semantic paths, empirical evaluation of this fixed-threshold approach is warranted but absent from the manuscript. Furthermore, since distant entities are generally less important than closer ones, justification for using a uniform threshold would be valuable.

4. For evaluation using LLMs as judges, it is recommended to include multiple state-of-the-art LLMs from different providers to reduce bias stemming from reliance on a single model.

---

> ### Author Response · Authors · 2025-11-21
> **Rebuttal to df73 (part 1/2)**
>
> Dear Reviewer df73,
>
> Thank you for this insightful question and for giving us the opportunity to clarify these points. Below are detailed responses to your comments:
>
> > ### **Regading Q1: Robustness to Entity Extraction Failures**
>
> 1. **When no entities can be extracted from the query**
>
> The retrieval process does not halt. LinearRAG automatically falls back to standard similarity-based dense passage retrieval using the original query embedding. This ensures the system remains robust even for entity-free or highly abstract questions.
>
> 2. **When extracted entities are ambiguous and cannot be disambiguated without the query’s full context**
>
> We recognized this as a potential false cause in graphRAG systems. **Therefore, we have taken this into consideration when designing our retrieval algorithm.** As described in the paper, during entity activation we compute the **query–sentence** relevance distribution ****$\sigma_q$, which encodes the **full query context**. This distribution is then used in Eq. (5) for semantic propagation. Consequently, ambiguously activated (noisy) entities receive low $\sigma_q$ support and do not effectively propagate relevance to their neighbours, while correctly activated entities dominate the propagation and prioritize the relevant intermediate entities.
>
> Furthermore, in the global importance aggregation stage (Sec. 3.2.2), passage importance scores are initialized with the direct query–passage similarity $sim(q, v)$. This additional signal again incorporates the complete query context and further suppresses passages reached only through ambiguous or incorrect entities.
>
> > ### **Regarding W1 and Q2: Clarification on Equation 5**
>
> Thank you for carefully checking the equations.
>
> We respectifully refer to Eq. (2) in the Sec 3.1, where we have defined the matrix $M$ as the adjacent matrix of the sentence-entity bipartite graph:
> $$
> M \in \{0,1\}^{|S| \times |E|}, \quad M_{s,e} = 1 \text{ if entity } e \text{ is mentioned in sentence } s.
> $$
> The concern about the dimension mismatch in Eq. 5 appears to stem from a typo in the submission draft. In the current updated manuscript that was uploaded for the discussion phase, Equation 5 has been corrected to:
> $$
> a_q^{(t)} = \max\ \Big( M^\top \big(\sigma_q \odot (M a_q^{(t-1)})\big), a_q^{(t-1)} \Big)
> $$
> The dimensions therefore align perfectly, and the final $\max$ is taken element-wise with $a_q^{(t-1)}$.
>
> For additional clarity, we have added a one-sentence reminder of $M$'s definition right before Equation 5.
>
> We sincerely appreciate you catching the outdated notation, which helped us improve the manuscript and enhance clarity. We have incorporated these revisions to the updated version.
>
> > ### **Regarding W2 and Q3: Effectiveness of Fixed-Threshold Pruning**
>
> The use of a fixed threshold $\delta$ is a deliberate and principled choice that aligns with the MAX-based propagation in Equation (5): unlike sum- or average-based message passing (where relevance naturally decays with distance), the element-wise maximum ensures that a strong semantic path preserves its full score even over many hops. Consequently, truly relevant distant entities can still exceed the threshold, while weak paths are aggressively suppressed regardless of distance.
>
> Therefore, an explicit distance-decaying threshold is unnecessary and potentially harmful: (1) it could prematurely discard valid long-range bridges or introduce more noise; (2) selecting the decay factor is difficult and may affect the generalization ability.
>
> That said, we fully agree that empirical validation would strengthen the paper. To demonstrate the necessity of dynamic pruning and the effectiveness of our fix threshold design, we compare LinearRAG against three alternative strategies: 1) w/o pruning, which disables pruning during entity activation process; and 2) w. dynamic threshold, which employs a threshold that decays with iterations, specifically, geometric decay (×0.5) and linear decay (−0.1).
>
> Table 1: Performance (%) and efficiency comparison  of different pruning strategies.
>
> | Method | Acc. | Avg. Retrieval Time (s) |
> | --- | --- | --- |
> | **LinearRAG (Full)** | **66.95** | **0.093** |
> | w.o pruning | 64.50 | 0.186 |
> | w. dynamic threshold (×0.5) | 65.15 | 0.123 |
> | w. dynamic threshold (−0.1) | 65.35 | 0.095 |
>
> The results confirm that the simple fixed threshold achieves the best accuracy/efficiency balance. Removing dynamic pruning or using decay-based strategies underperforms the fixed approach, as both increase retrieval time and harm the accuracy of generation.
>
> Thanks for your suggestion on extending our discussions on this point, we have included them into Table 3 in the manuscripts.

---

> ### Author Response · Authors · 2025-11-21
> **Rebuttal to df73 (part 2/2)**
>
> > ### **Regarding W3 and Q4: Evaluation with Multiple LLMs**
>
>
> We fully agree that using multiple LLM evaluators can help reduce potential evaluation bias. To address this concern, we conduct additional experiments using three different LLMs as evaluators: GPT-4o-mini, DeepSeek-Chat, and Claude-Haiku-4.5. We compare LinearRAG against the most advanced baselines across four benchmark datasets. The results are presented below:
>
> Table 2: GPT-Acc (%) of different methods across LLM evaluators.
>
> | Method | LLM  | HotpotQA | 2Wiki | MuSiQue | Medical |
> | --- | --- | --- | --- | --- | --- |
> | HippoRAG2 | gpt-4o-mini | 64.30 | 55.00 | 35.00 | 60.77 |
> |  | deepseek-chat | 69.70 | 62.30 | 38.80 | 64.40 |
> |  | claude-haiku-4.5 | 63.60 | 58.50 | 35.60 | 62.12 |
> | GFMRAG | gpt-4o-mini | 65.60 | 59.60 | 34.60 | 56.07 |
> |  | deepseek-chat | 70.70 | 68.10 | 36.50 | 60.58 |
> |  | claude-haiku-4.5 | 63.60 | 64.80 | 34.30 | 56.84 |
> | RAPTOR | gpt-4o-mini | 58.30 | 42.10 | 27.40 | 55.75 |
> |  | deepseek-chat | 62.90 | 48.60 | 30.40 | 58.39 |
> |  | claude-haiku-4.5 | 57.50 | 44.90 | 27.00 | 56.21 |
> | **LinearRAG** | gpt-4o-mini | **66.50** | **63.70** | **37.00** | **63.72** |
> |  | deepseek-chat | **72.60** | **70.20** | **39.90** | **66.54** |
> |  | claude-haiku-4.5 | **65.90** | **66.20** | **36.90** | **64.11** |
>
> The results show that LinearRAG consistently outperforms all baseline methods across all three LLM evaluators, maintaining its advantage despite different scoring patterns among evaluators. This result has been included as Table 4 in the manuscript.

---

> > ### Comment · Reviewer_df73 · 2025-11-25
> >
> > Thank the authors for their comprehensive responses to my comments and my concerns have been addressed.

---

> > > ### Author Response · Authors · 2025-11-27
> > > **Thank you**
> > >
> > > Dear Reviewer df73,
> > >
> > > Thanks a lot for your recognition of our work and for providing such thorough and insightful feedback. Your comments and suggestions are invaluable in helping us improve the manuscript. Following your suggestions, we have made the corresponding revisions in the updated version.
> > >
> > > Best regards,
> > >
> > > Authors of Paper9474

---

### Official Review · Reviewer_5Por · 2025-10-30

**Soundness:** 3
**Presentation:** 3
**Contribution:** 3
**Rating:** 6
**Confidence:** 4

**Summary:**

This paper proposes a new framework for RAG, LinearRAG, =which overcomes the limitations of existing GraphRAG systems. The proposed LinearRAG replaces complex relation modeling with a relation-free hierarchical Tri-Graph, built using lightweight entity extraction and semantic linking. This design scales linearly with corpus size and avoids extra token consumption. The retrieval process follows a two-stage mechanism: First, local semantic bridging activates relevant entities beyond literal matches through semantic propagation. Second, global importance aggregation applies personalized PageRank to rank passages holistically.

Experiments on HotpotQA, 2WikiMultiHopQA, MuSiQue, and a Medical benchmark show that LinearRAG consistently outperforms both vanilla RAG and advanced GraphRAG methods in retrieval quality, generation accuracy, and efficiency. It achieves strong multi-hop reasoning performance with minimal computational overhead.

LinearRAG offers a practical and scalable solution for integrating structured retrieval with large-scale text corpora, avoiding the instability of relation extraction and achieving linear scalability in both construction and retrieval.

**Strengths:**

- This paper proposes a efficient and lightweight graph construction method, Tri-Graph, which avoids error-prone relation extraction in GraphRAG, maintains linear scalability, and achieves 0 token consumption in retrieval.
- This paper proposes a novel GraphRAG framework that combine local semantic bridging and global importance aggregation, whose effectiveness is valided through extensive experiments.
- This paper conducted extensive experiments to validate the effectiveness of LinearRAG, including sufficient main experiments against GraphRAG and vanilla RAG.

**Weaknesses:**

- The ablation study seems coarse-grained. For example, when constructing Tri-Graph, the current method simultaneously incorporates co-occurrence relationships between entities and both passages and sentences. How would the final performance be affected if only passages, only sentences, or paragraphs instead?
- The main experiment only compared with vanilla RAG. What about comparisons with some advanced RAG methods, such as LightRAG, Lexical Diversity-aware RAG, and others?

**Questions:**

- The token-free graph construction process appears similar to the context-graph construction in the paper *"Synthesize-on-graph: ..."*, as both utilize the position relationships between entities and texts as graph relationships. Could you please explain the key distinctions of your method?
- Other questions refer to Weaknesses.

---

> ### Author Response · Authors · 2025-11-21
> **Rebuttal to 5Por**
>
> Dear Reviewer 5Por,
>
> Thank you for your expertise comments. Below are detailed responses to your comments:
>
> > ### **Regarding W1: Ablation Study**
>
> To address the reviewer's concern regarding the contribution of each node granularity in the Tri-Graph. We evaluate two simplified graph structures: (1) a graph containing only the passage-entity subgraph, and (2) a graph with only the sentence-entity subgraph.
>
> Table 1: Ablation results (%) of different graph structures. The reported values represent the average of GPT-Acc. and Contain-Acc.
>
> | Method | HotpotQA | 2Wiki | MuSiQue | Medical |
> | --- | --- | --- | --- | --- |
> | **LinearRAG (Full)** | **65.40** | **66.95** | **35.45** | **63.72** |
> | passage-entity only | 63.15 | 64.40 | 31.65 | 61.69 |
> | sentence-entity only | 63.35 | 64.20 | 32.05 | 61.73 |
>
> The experiments show that accuracy degrades on simplified graph structures, with consistent performance drops of 2.0-3.8% when removing either node type. This demonstrates that both granularities play complementary roles in our two-stage retrieval: sentence nodes enable precise entity activation through fine-grained semantic bridging, while passage nodes support global importance aggregation for retrieving coherent contexts.
>
> Regarding the suggestion to use "paragraph" granularity, we note that paragraphs lack a universal definition across diverse corpora, as not all datasets (e.g., academic papers, technical documents) preserve natural paragraph boundaries, which limits the generalizability of such an approach. In contrast, sentence segmentation via punctuation is universally applicable. Furthermore, replacing sentence nodes with paragraph nodes would introduce excessive noise during entity activation, as paragraphs contain 5-8 entities on average compared to 2-3 entities per sentence.
>
> > ### **Regarding W2: Advanced RAG Baselines**
>
> In addition to the vanilla RAG baseline, we compare LinearRAG with several advanced RAG methods. The results are summarized in Table 2.
>
> Table 2: Performance (%) of Baseline Methods and LinearRAG on Four Benchmark Dataset. The reported values represent the average of GPT-Acc. and Contain-Acc.
>
> | Method | HotpotQA | 2Wiki | MuSiQue | Medical |
> | --- | --- | --- | --- | --- |
> | ReRankRAG | 55.70 | 47.30 | 29.25 | 55.43 |
> | IRCoT | 58.45 | 47.55 | 28.70 | 48.69 |
> | LightRAG | 59.80 | 47.10 | 28.00 | 54.36 |
> | **LinearRAG** | **65.40** | **63.7** | **35.45** | **63.72** |
>
> As shown in Table 2, LinearRAG consistently outperforms all advanced baseline methods across all four datasets. Notably, we have added these baselines into the Table 1 in the manuscripts.
>
> > ### **Regading Q1: Comparison with Synthesize-on-Graph**
>
> While both LinearRAG and Synthesize-on-Graph (SOG) leverage entity-text relationships in graph construction, they differ fundamentally in task objectives, graph architecture, and retrieval mechanism:
>
> - **Task Objectives:** SOG focuses on training data synthesis to generate diverse training samples, while LinearRAG targets retrieval-augmented generation for answering multi-hop queries.
> - **Graph Architecture:** SOG constructs an entity co-occurrence graph where entities are connected if they appear in the same passage, without explicit passage nodes. In contrast, LinearRAG builds a three-layer "Tri-Graph" with entity, sentence, and passage nodes.
> - **Retrieval Mechanism:** SOG uses BFS traversal on entity co-occurrence graphs to construct knowledge paths for synthesis. LinearRAG employs two-stage retrieval: entity activation via sentence-level semantic bridging, followed by passage retrieval via Personalized PageRank.

---

### Official Review · Reviewer_qxoV · 2025-10-30

**Soundness:** 3
**Presentation:** 3
**Contribution:** 3
**Rating:** 6
**Confidence:** 4

**Summary:**

This paper proposes LinearRAG, a novel GraphRAG framework designed to address key challenges in existing methods: the difficulty and noise associated with knowledge graph construction, and low retrieval efficiency. The core innovation is constructing a three-layer hierarchical graph (Tri-Graph) comprising entities, sentences, and paragraphs. During retrieval, the model first activates query-relevant entities via local semantic propagation and then uses these as seeds in a Personalized PageRank algorithm on a paragraph-entity bipartite graph to aggregate global importance for retrieval in a single pass. Both the time and space complexity scale linearly with the corpus size, and the entire indexing/retrieval process requires zero LLM API calls. Experimental results on four public datasets demonstrate that LinearRAG significantly outperforms existing GraphRAG methods.

**Strengths:**

1. The paper presents a strong, well-argued case that relation extraction errors are a primary bottleneck for GraphRAG performance. The decision to "de-relationalize" the graph and defer relation understanding to the LLM during generation is a conceptually clear and impactful design choice.

2. The proposed Tri-Graph structure is a novel and potentially groundbreaking simplification of the GraphRAG paradigm. The retrieval mechanism, leveraging sparse matrix operations for semantic propagation and PPR, is computationally elegant.

3. The framework's linear scalability, facilitated by update-friendly appends and the complete absence of LLM calls in indexing/retrieval, is a major strength. The demonstrated linear growth trend on a million-scale corpus underscores its high practical value for real-world deployment.

4. The paper provides an exceptionally thorough evaluation, going beyond standard QA accuracy to include critical efficiency metrics like token cost and latency. This holistic comparison makes the results highly persuasive and relevant for computation-budget-sensitive scenarios.

**Weaknesses:**

1. The Tri-Graph merges entity mentions based on string matching, which ignores polysemy (different entities with the same name). This is a significant limitation, as it can introduce erroneous co-occurrence edges, potentially leading to the retrieval of irrelevant paragraphs. While high NER accuracy is claimed, the paper does not quantify the impact of this inherent ambiguity on retrieval performance, leaving a major potential source of error unanalyzed.

2. Using binary weights (0/1) for sentence-entity edges fails to capture the varying semantic importance of different entities within a sentence. This uniform weighting strategy is a simplification that may amplify the influence of "noisy" or peripheral entities during the propagation step, potentially harming result relevance and interpretability.

3. The parameters, particularly the activation threshold and the trade-off coefficient, are noted to be highly sensitive. The paper lacks an analysis establishing a functional relationship between these parameters and query characteristics or graph properties.

**Questions:**

Could the framework benefit from more informative weights for sentence-entity edges?

---

> ### Author Response · Authors · 2025-11-21
> **Rebuttal to qxoV**
>
> Dear Reviewer qxoV，
>
> Thank you for your recognition of our work and for providing such thorough and insightful feedback. Your comments and suggestions are invaluable in helping us improve the quality and clarity of our work.
>
> > ### **Regarding W1: Polysemy in Entity Matching**
>
> We acknowledge that polysemy is a common challenge in information retrieval system. When constructing indexing graphs with NER-extracted entities, different passages may be connected through the entities sharing same name (e.g. fruit $Apple$ and Company $Apple$ ), leading to potential false retrieval. However, our design can largely alleviate this issue:
>
> First, as you have noticed and mentioned in your first strength point, our design - deferring relation understanding to the LLM during generation, can largely alleviate this issue. As it leverage the LLM's semantic understanding ability to distinguish the conceptual differences between different entities by analyzing the original context, filtering the noise information from irrelevant passages.  This is a unique benefit of LinearRAG's relation-free mechanism, compared with GraphRAG algorithms that abstracting passages into relational KGs and matching abstracted relation path.
>
> Second, LinearRAG employs sentence-level semantic bridging during entity activation, judging entity relevance through the context of sentences. Homonymous entities with different meanings are naturally distinguished by the semantic differences in their surrounding sentences.
>
> Furthermore, LinearRAG can naturally integrate 'self-reflection' and 'query-rewrite' strategies as post refinement for the RAG process.
>
>
> > ### **Regarding W2 and Q1: Binary vs. Weighted Sentence-Entity Edges**
>
> We deliberately adopted binary sentence–entity edges to preserve LinearRAG’s efficiency guarantees: fully LLM-free indexing and linear-time construction during graph building. Introducing richer precomputed weights (e.g., TF-IDF, LLM-based scores) would require additional computation and thus break these guarantees.
>
> That said, we fully recognize the reviewer’s point that uniform weighting is a simplification that could, in principle, allow peripheral or noisy entities to participate in propagation. We address this concern from two complementary angles:
>
> **1. Query-agnostic precomputed weights can actually hurt recall**
>
> Assigning fixed importance weights to entities without knowledge of the incoming query risks systematically down-weighting “peripheral” entities that are, for that particular query, critical intermediate anchors in multi-hop chains. Binary weighting, by contrast, treats all mentioned entities democratically and therefore preserves the possibility of activating less-frequent but query-relevant bridges. This design choice favours recall (especially on multi-hop benchmarks) over premature filtering of potentially useful entities.
>
> **2. Multiple built-in mechanisms already suppress noisy/peripheral entities at query time**
>
> - **Granularity of sentences**: Unlike passages (which often contain 8–15 entities), a typical sentence contains only 2–3 entities. This naturally limits the fan-out of noisy entities and makes binary weighting far less harmful than it would be on coarser passage–entity graphs.
> - **Dynamic pruning (Sec. 3.2.1)**: Entities must exceed a relevance threshold $\delta$ to be carried forward, further filtering peripheral or noisy entities.
>
> > ### **Regarding W3: Parameter Analysis**
>
> Contrary to the reviewer’s impression that the parameters are “highly sensitive”, the existing sensitivity experiments in the submitted manuscript (Appendix E.2) clearly show that both $\delta$ and $\lambda$ are robust on 2WikiMultiHopQA:
>
> Table 1: The impact of threshold $\delta$ on 2WikiMultiHopQA dataset. Acc. means the average scores of GPT-Acc. and Contain-Acc.
>
> | δ | 0.2 | 0.4 | 0.6 | 0.8 |
> | --- | --- | --- | --- | --- |
> | Acc. | 66.75 | **66.95** | 66.30 | 64.65 |
>
> Table 2: The impact of threshold $\lambda$ on 2WikiMultiHopQA dataset. Acc. means the average scores of GPT-Acc. and Contain-Acc.
>
> | λ | 0.01 | 0.05 | 0.5 | 1.0 |
> | --- | --- | --- | --- | --- |
> | Acc. | 65.80 | **66.95** | 65.80 | 65.95 |
>
> Performance remains within about 2% below the peak across a wide range of threshold values ($\delta$ ∈ [0.2, 0.8]), while still outperforming all baseline methods. Since $\delta$ = 0.4 achieves the optimal performance, we adopt it as the default setting. And the trade-off coefficient should be a small value so that entity information serves as the primary component while DPR similarity score acts as a supplementary enhancement.

---

### Official Review · Reviewer_oM4K · 2025-11-01

**Soundness:** 3
**Presentation:** 3
**Contribution:** 3
**Rating:** 6
**Confidence:** 3

**Summary:**

The paper proposes LinearRAG, an efficient GraphRAG framework that constructs a relation-free hierarchical graph (Tri-Graph) using lightweight entity extraction and semantic linking over entities, sentences, and passages. For retrieval, LinearRAG adopts a two-stage strategy: (1) local semantic bridging to activate contextually relevant entities beyond literal matches, and (2) global importance aggregation via personalized PageRank to retrieve salient passages. This construction scales linearly with corpus size and incurs no extra token consumption, providing economical and reliable indexing for large corpora. Experiments on four benchmarks show consistent improvements over strong baselines in retrieval quality, generation accuracy, and scalability.

**Strengths:**

S1: The approach fits production constraints and integrates with standard embedding + reranking pipelines.

S2: The linear retrieval design emphasizes bounded expansions and pruning, offering predictable runtime and memory behavior on large graphs.

S3: The method encourages simple deployment: light graph construction, single-pass adjacency, and top-k selection, reducing system complexity compared to full GraphRAG stacks.

**Weaknesses:**

W1: Linear constraints risk under-retrieving long-range dependencies; multi-hop reasoning and compositional queries may suffer without fallback mechanisms.

W2: Limited theoretical analysis (e.g., recall/latency trade-offs, worst-case behaviors) could make the contribution primarily engineering rather than principled.

**Questions:**

Q1: What exact linear retrieval procedure is used (single-pass BFS-like scan, bounded-degree expansion, heuristic queue ordering)? How are thresholds chosen and adapted?

Q2: How are node and edge attributes computed (embedding similarity, co-occurrence, LLM scoring)? Are edge weights normalized and thresholded consistently?

Q3: What fallback exists for multi-hop or compositional queries requiring evidence aggregation? Can limited expansions be temporarily enabled without breaking linearity guarantees?

Q4: Are there systematic trade-off curves (recall vs. latency/memory) across different graph sizes and domains? How robust are results to parameter changes?

Q5: Do comparisons include equal-resource settings against beam search or greedy path selection to isolate algorithmic gains from compute differences?

---

> ### Author Response · Authors · 2025-11-21
> **Rebuttal to oM4K (part 1/2)**
>
> Dear Reviewer oM4K，
>
> Thank you for recognizing our work and for providing such thorough and insightful feedback. Below are detailed responses to your comments and suggestions:
>
> > ### **Regarding W1 and Q3: Long-Range Dependency Resolution**
>
>
> We fully agree that multi-hop and compositional queries are challenging for any RAG system, and this is exactly one of our design considerations in this work.
>
> We respectfully refer to Sec. 3.2 in the manuscripts, where our two-stage design (local semantic bridging → global importance aggregation) is explicitly built **to capture long-range dependencies** without explicit relation triples:
>
> - The first stage can be understood as a progressive clue-following mechanism that performs iterative entity activation on the sentence–entity bipartite graph to explore an $n$-hop subgraph that potentially anchors the relevant context passages.
> - The second stage scores each passage based on two factors: the relevance between activated entities and passages, and the semantic similarity between the query and passages. Personalized PageRank is then applied on the passage–entity bipartite graph to propagate importance from activated entities and rank passages globally.
>
> These designs naturally capture long-range dependencies during retrieval and empirically demonstrate superior performance on four multi-hop benchmarks. The case study refers to Table 11 in the manuscripts, showing how this mechanism works with a concrete example.
>
> > ### **Regarding W2 and Q4: Recall-lantency trade-off**
>
>
> Due to LinearRAG's linear design, the tunable hyperparameters $\delta$ and $\lambda$ have no impact on retrieval latency. They only prune similarity thresholds and do not change the amount of computation. The only parameter that trades recall for latency is the number of iterations in Eq. (5). This number is conceptually the maximum hop distance and is therefore bounded by the question's hop count. In practice, ≤4 suffices for all questions. Each iteration is a few vectorized similarity computations that can be executed very efficiently.  Following your suggestion, we have provided a comprehensive theoretical analysis of LinearRAG in Appendix D.
>
> Besides, we investigate the impact of graph size and different domains on retrieval efficiency. With the default of 3 iterations, the average retrieval latencies across four datasets of varying scales on a 4090 GPU are presented in Table 1. Specifically, we include the Novel dataset from GraphRAG-Bench to enhance the diversity of domains.
>
> Table 1. Graph size and retrieval latency statistics across datasets.
>
> | Dataset |  Graph Size | Avg. Retrieval Time (s) |
> | --- | --- | --- |
> | Medical | 202,126 | 0.081 |
> | 2Wiki | 517,370 | 0.093 |
> | Novel | 1,027,013 | 0.102 |
> | HotpotQA | 1,062,480 | 0.107 |
>
> Given that this step is both theoretically and practically very fast, we believe it is reasonable to set a small number (3-4) of iterations to obtain substantial accuracy gain without any meaningful trade-off in latency. And we have included them into Appendix E.5 in the manuscript.
>
> > ### **Regarding Q1: Linear Retrieval Process and Threshold Selection**
>
> As above, the retrieval includes an entity activation stage followed by a global importance computation stage. The entity activation stage is essentially a BFS-algorithm, although we formulated and implemented it as a message passing process to enable an accelerated process with vectorized computation.
>
> Regarding the threshold $\delta$ selection, we refer to the sensitivity analysis presented in Appendix E.2 of the manuscript.
>
> Table 2: The impact of threshold $\delta$ on the 2WikiMultiHopQA dataset. Acc. means the average scores of GPT-Acc. and Contain-Acc.
>
> | δ | 0.2 | 0.4 | 0.6 | 0.8 |
> | --- | --- | --- | --- | --- |
> | Acc. | 66.75 | **66.95** | 66.30 | 64.65 |
>
> As shown in Table 2, performance remains within about 2% below the peak across a wide range of threshold values ($\delta$ ∈ [0.2, 0.8]), while still outperforming all baseline methods. Since $\delta$ = 0.4 achieves the optimal performance, we adopt it as the default setting.

---

> ### Author Response · Authors · 2025-11-21
> **Rebuttal to oM4K (part 2/2)**
>
> > ### **Regarding Q2: Node/Edge Attribute Computation**
>
> We adopt binary sentence-entity and passage-entity edges to preserve LinearRAG's efficiency advantages: entirely LLM-free indexing and linear time complexity during graph construction.
>
> Beyond efficiency, we think that query-agnostic fixed weights can actually hurt recall. Assigning fixed importance weights without knowledge of the incoming query risks systematically down-weighting "peripheral" entities that are, for that particular query, critical intermediate anchors in multi-hop chains. Binary weighting, by contrast, treats all mentioned entities democratically and preserves the possibility of activating less-frequent but query-relevant bridges. This design choice favors recall (especially on multi-hop benchmarks) over premature filtering of potentially useful entities.
>
> > ### **Regarding Q5: Comparison with Beam and Greedy Search**
>
> To isolate algorithmic gains from computational differences, we implemented beam search and greedy path selection strategies within our framework. Specifically, we modified the relevant entity activation stage by selecting either the top-1 (greedy) or top-3 (beam) most similar sentence nodes for propagation at each step.
>
> Table 3: Accuracy and efficiency comparison with beam search and greedy search on 2WikiMultiHopQA.
>
> | Method | Acc. | Avg. Retrieval Time (s) |
> | --- | --- | --- |
> | LinearRAG | 66.95 | 0.093 |
> | Beam Search | 65.90 | 0.091 |
> | Greedy Search | 64.85 | 0.087 |
>
> As shown in Table 3, all variants operate under similar retrieval times (0.087–0.093s), yet LinearRAG achieves 1–2% higher accuracy. This improvement stems from its flexible entity activation process, which explores multiple paths simultaneously and can recover from suboptimal initial matches, whereas the simplified variants tend to get stuck on early decisions. Notably, we have added these results to Table 10 in the manuscripts.

---

### Author Response · Authors · 2025-12-02
**General Response**

Dear Area Chairs and Reviewers,

We sincerely thank you for your time, effort, and invaluable feedback during the author-reviewer discussion phase. Your insights have been crucial in helping us refine and improve our work. To facilitate the next stage of review and discussion, we would like to summarize our paper, the main concerns, and our responses.

## **Overall Assessments**

Our initial scores were **(6, 6, 6, 6)** with reviewer confidence ratings of (3, 4, 4, 5). And these scores remained consistently positive until the appearance of the major vulnerability on OpenReview. Specifically, Reviewer df73 explicitly stated that our responses have addressed the concerns.

> Reviewer df73: Thank the authors for their comprehensive responses to my comments, and my concerns have been addressed.

## **Summary of Our Work**

Graph-based RAG (GraphRAG) has been proposed to capture relational structures, enabling more comprehensive retrieval on multi-hop reasoning tasks, but extracting concise and accurate relational triples is **computationally expensive and linguistically challenging**. In this paper, we propose a **relation-free framework**, Linear Graph-based Retrieval-Augmented Generation (LinearRAG), that enables efficient, reliable graph construction and precise passage retrieval. Our main contributions are as follows:

- We identify the fundamental bottleneck in GraphRAG and propose LinearRAG, addressing a timely and critical challenge faced by both industry and academia.
- We introduce Tri-Graph, a lightweight graph construction method achieving linear scalability with zero LLM token costs.
- We design a two-stage retrieval mechanism integrating local semantic bridging with global importance aggregation with minimal latency and zero LLM token consumption.
- We provide comprehensive experiments across four datasets against 17 baselines.

## **Main Concerns and Our Responses**

- **Hyperparameter Analysis:** Reviewer oM4K and qxoV suggested adding an analysis of the parameter sensitivity. We investigate the impact of the threshold $\delta$ and trade-off coefficient $\lambda$ in LinearRAG. The results are presented in Figure 5 (Appendix E.2). Further, following the suggestion from Reviewer df73, we compared fixed threshold with dynamic decay thresholds. The results are presented in Table 3 (Section 4.4)
- **Impact of Edge Weights:** Reviewer oM4K and qxoV raised questions about the details of edge weights. We have added a detailed explanation and further clarified the advantages of binary sentence-entity edges. The complete discussion is provided in Appendix G.3.
- **Comparison with Traditional RAG Baselines:** Reviewer 5Por suggested comparing with more traditional RAG baselines. We conducted experiments with ReRankRAG, IRCoT, and LightRAG across four benchmarks. The results are presented in Table 1 (Section 4.2).
- **Efficiency Analysis:**  Reviewer oM4K suggested that we supplement the analysis on the impact of graph size/domain on retrieval efficiency. We analyzed average retrieval time across different graph sizes/domains in Table 9 (Appendix E.5) and provide a theoretical analysis in Appendix D.

## **Conclusion**

Thank you again for your time and expertise throughout this review process. We deeply appreciate your expertise and recognition of our work. Each point raised has given us new perspectives to consider. Hope that this summary could facilitate the next stage of review and discussion.

---

### Meta-Review · Area_Chair_yybW · 2025-12-17

**Summary:**

This paper proposes Linear Graph-based Retrieval-Augmented Generation (LinearRAG), a relation-free framework that enables efficient and reliable graph construction while supporting precise passage retrieval. The concerns of revirwers mainly lie in the experimetns. The concerns of reviewers have been solved during rebuttal. I think these additional results should be added in the next version.

**Reviewer Concerns:**

The reviewers have proposed some concerns:
1. Linear constraints may not be able to fully support multi-hop reasoning and combinatorial queries. They lack clear fallback or expansion mechanisms, and there is a risk of insufficient long-range dependency recall.
2. The lack of systematic analysis of trade-offs such as recall rate – latency – memory and worst-case behavior makes the method appear more like an engineering design rather than a principled approach.
3. The ablation granularity is coarse, and it only compares with vanilla RAG, lacking fair comparisons with stronger RAG methods and under equal computing power settings.
4. Key parameters (such as activation threshold, trade-off coefficient) are highly sensitive to performance, but their relationship with query features or graph structure has not been analyzed.
During rebuttal, these concerns have been solved.

**Reviewer Scores:**

I believe these scores remain unchanged.

---

### Decision · Program_Chairs · 2026-01-26

Accept (Poster)